# Making Reliable and Flexible Decisions in Long-tailed Classification

## Abstract

Long-tailed classification is challenging due to its heavy imbalance in class probabilities. While existing methods often focus on overall accuracy or accuracy for tail classes, they overlook a critical aspect: certain types of errors can carry greater risks than others in real-world long-tailed problems. For example, misclassifying patients (a tail class) as healthy individuals (a head class) entails far more serious consequences than the reverse scenario. To address this critical issue, we introduce Making **R**eliable and **F**lexible **D**ecisions in **L**ong-tailed **C**lassification (RF-DLC), a novel method aimed at ensuring reliable predictions in long-tailed problems. Leveraging Bayesian Decision Theory, we introduce an integrated gain to seamlessly combine long-tailed data distribution and the decision-making procedure. We further propose an efficient variational optimization strategy for the decision risk objective. Our method adapts readily to diverse utility matrices, which can be designed for specific tasks, ensuring its flexibility for different problem settings. In empirical evaluation, we design a new metric, False Head Rate, to quantify tail-sensitivity risk, and conduct comprehensive experiments on multiple real-world tasks, including classification, uncertainty estimations, and ablation studies, to demonstrate the reliability and flexibility of our method.

## 1 Introduction

Real-world categorical data is often long-tailed distributed, where the data distribution is biased towards a few "head" classes and the "tail" classes have much fewer samples (Reed, 2001; Lin et al., 2014; Van Horn & Perona, 2017; Krishna et al., 2017; Liu et al., 2019b; Wang et al., 2020; Li et al., 2022). The long-tailed problem primarily stems from the inherent biases in the data collection process, which are challenging to avoid. Models trained on long-tailed data using conventional methods often exhibit notable performance drops compared to those trained on balanced datasets (Wang et al., 2022).

Existing approaches to long-tailed classification usually focus on improving overall accuracy or accuracy for tail classes, such as re-weighting loss (Lin et al., 2017; Cao et al., 2019; Cui et al., 2019; Wu et al., 2020), logit adjustment (Menon et al., 2020; Ren et al., 2020; Hong et al., 2021), and knowledge transfer (Liu et al., 2019b; Xiang et al., 2020). However, in practical applications, the ultimate goal is often not accuracy but making optimal decisions. Existing methods on long-tailed classification assume that mispredictions between any pair of classes are equally risky, which is often violated in real-world tasks. The probability of misclassifying tailed samples as head samples is very high in existing methods. For example, in disease detection where healthy individuals belong to head classes and patients belong to tail classes, the risk of classifying patients as healthy individuals is significantly larger than the reverse scenario (Yang et al., 2022). In autonomous driving, mispredicting tail classes like pedestrians or cyclists as head classes like vehicles will significantly increase the risk of accidents and injuries (Carranza-García et al., 2021).

To enable reliable long-tailed classification and simultaneously integrate decision-making, we propose Making **R**eliable and **F**lexible **D**ecisions in **L**ong-tailed **C**lassification (RF-DLC), a general learning framework aimed at reliable predictions[1] on diverse realistic long-tailed problems. Specifically, we introduce the integrated

---

[1]The words "prediction" and "decision" are used interchangeably throughout this paper.

gain from *Bayesian Decision Theory* (Robert et al., 2007; Berger, 2013), an important branch of cost-sensitive learning, which allows us to seamlessly incorporate decision risk and long-tailed data distribution in a single objective. While cost-sensitive learning has been applied to standard classification (Elkan, 2001; Chung et al., 2016; Shu et al., 2019), its application to long-tailed classification remains unexplored. We also propose a variational optimization strategy to efficiently maximize the integrated gain w.r.t. model parameters. Leveraging the notion of "utility matrix", our method is flexible to many real-world tasks with different types of risks. In our empirical evaluations, we design a new metric, False Head Rate (FHR), to quantity the mispredictions from tail classes to head classes—a common source of high risk in long-tailed classification (Sengupta et al., 2016; Rahman et al., 2021; Yang et al., 2022).

The main contributions of this paper are summarized as follows:

- RF-DLC is the first to consider decision-making in long-tailed classification. Built upon Bayesian Decision Theory, RF-DLC enables principled and reliable predictions on long-tailed data.

- RF-DLC introduces a new objective called integrated gain, an efficient variational optimization strategy, and several utility matrices tailored for different long-tailed scenarios. These techniques are naturally derived from Bayesian Decision Theory, ensuring that they are integral to the model's function rather than add-on tricks.

- RF-DLC is flexible and can be adapted to various tasks with diverse metrics. Users can adopt our method to many specific fields by re-designing utility matrices.

- We conduct comprehensive experiments to demonstrate that RF-DLC significantly improves decision-making while maintaining or improving traditional metrics such as accuracy and calibration.[2]

## 2 Related Works

**Long-tailed Classification.** Previous methods mainly tackle long-tailed classification from the following aspects: i) adjusting data distribution to obtain balanced datasets, including over-sampling (Han et al., 2005), under-sampling (Liu et al., 2008) and data augmentation (Chu et al., 2020; Kim et al., 2020; Liu et al., 2020)[3]; ii) re-balancing the importance of different classes in loss functions, including re-weighting loss (Lin et al., 2017; Mahajan et al., 2018; Cao et al., 2019; Cui et al., 2019; Menon et al., 2020; Wu et al., 2020) and logit adjustment (Menon et al., 2020; Ren et al., 2020; Hong et al., 2021); iii) applying heterogeneous model architectures to handle head and tail samples in different ways, including OLTR (Liu et al., 2019b), LFME (Xiang et al., 2020), RIDE (Wang et al., 2020), TLC (Li et al., 2022) and SRepr (Nam et al., 2023). Other attempts explore broader long-tailed classifications, including non-uniform testing distributions (Zhang et al., 2022), outlier samples (Wang et al., 2022; Bai et al., 2022), and partial-labeled datasets (Hong et al., 2022). Our method is significantly different from existing methods by targeting decision-making and asymmetric misprediction risks, which are important problems in realistic long-tailed data.

**Bayesian Decision Theory and Cost-sensitive Learning.** Robert et al. (2007); Berger (2013) have comprehensively introduced Bayesian Decision Theory, which can be adapted to many different tasks by the design of utility functions. For its usage in modern machine learning, multiple ways have been explored: i) loss-calibrated variational inference, including Loss-calibrated EM (Lacoste-Julien et al., 2011), LCVB (Jaiswal et al., 2020) and variational inference on continuous utilities (Kuśmierczyk et al., 2019); ii) loss-calibrated expectation propagation (Morais & Pillow, 2022). Bayesian Decision Theory is one of the main methodologies to solve cost-sensitive learning (Elkan, 2001; Ling & Sheng, 2008; Chung et al., 2016), which focuses on decision-making under heterogeneous misprediction costs. Our method is built upon Bayesian Decision Theory and introduces the integrated gain to consider the long-tailed data distribution in the decision-making procedure, which has not been explored in previous works.

---

[2]The code will be publicly available upon acceptance.
[3]These methods fall behind the SOTA of long-tailed classification for many years. Therefore, we did include them in the experiments.

## 3    Background

This section provides the necessary background knowledge on long-tailed classification and Bayesian Decision Theory for understanding our method.

**Long-tailed Distribution.**    In long-tailed categorical data, the training and testing sets follow different distributions (Moreno-Torres et al., 2012; Cui et al., 2019; Liu et al., 2019b). Specifically, the training data is distributed in a descending manner over categories in terms of class probability:

$$p(y_1 = k_1) \geq p(y_2 = k_2), \quad \text{if } k_1 \leq k_2, \tag{1}$$

while the testing data is assumed to be uniform over categories: $p(y_1 = k_1) = p(y_2 = k_2)$ for all pairs of $(k_1, k_2)$. The distributional shift between training and testing data makes long-tailed problems challenging.

**Bayesian Decision Theory.**    To make optimal decisions in diverse problem settings, Bayesian Decision Theory provides a principled framework (Robert et al., 2007; Berger, 2013). In the supervised setting, for a dataset $\mathcal{D} = \{\boldsymbol{X}, \boldsymbol{Y}\} = \{(\boldsymbol{x}_i, y_i)\}_{i=1}^N$ and a $\boldsymbol{\theta}$-parameterized model, we denote the likelihood and prior to be $\prod_{i=1}^N p(y_i|\boldsymbol{x}_i, \boldsymbol{\theta})$ and $p(\boldsymbol{\theta})$ respectively. The posterior distribution is then $p(\boldsymbol{\theta}|\mathcal{D}) = p(\boldsymbol{\theta}) \prod_{i=1}^N p(y_i|\boldsymbol{x}_i, \boldsymbol{\theta}) / \prod_{i=1}^N p(y_i|\boldsymbol{x}_i)$. We further assume a *decision gain* $g(d|\boldsymbol{x}, \theta)$ to quantify the utility gained by choosing decision $d$ for input $\boldsymbol{x}$ when the model with parameters $\boldsymbol{\theta}$ controls the mapping from $\boldsymbol{x}$ to $y$. Subsequently, the *posterior expected gain* for an input $\boldsymbol{x}$ is defined as

$$G(d|\boldsymbol{x}, \mathcal{D}) := \mathbb{E}_{\boldsymbol{\theta} \sim p(\boldsymbol{\theta}|\mathcal{D})} g(d|\boldsymbol{x}, \boldsymbol{\theta}), \tag{2}$$

where we average over all possible models weighted by their posterior probabilities. Previous works (Lacoste-Julien et al., 2011; Cobb et al., 2018) maximize this objective w.r.t. model parameters to obtain a decision-calibrated posterior $q(\boldsymbol{\theta})$.

## 4    Methodology

In this section, we introduce RF-DLC, a method aimed at optimal decision-making for long-tailed data, grounded in Bayesian Decision Theory. The conventional posterior expected gain in Eq. 2 implicitly assumes that both training and testing data share the same distribution, which is violated in long-tailed problems. To address this challenge, we adopt the concept of the *integrated gain* from Bayesian Decision Theory, which incorporates the data distribution into the posterior expected gain as follows:

$$G(\boldsymbol{d}) := \mathbb{E}_{(\boldsymbol{x}_1, y_1), \dots, (\boldsymbol{x}_N, y_N) \sim p(\boldsymbol{x}, y)} G(\boldsymbol{d}|\boldsymbol{X}, \mathcal{D}) = \mathbb{E}_{(\boldsymbol{x}_1, y_1), \dots, (\boldsymbol{x}_N, y_N) \sim p(\boldsymbol{x}, y)} \mathbb{E}_{\boldsymbol{\theta} \sim p(\boldsymbol{\theta}|\mathcal{D})} G(\boldsymbol{d}|\boldsymbol{X}, \boldsymbol{\theta}), \tag{3}$$

where $G(\boldsymbol{d}|\boldsymbol{X}, \boldsymbol{\theta}) = \prod_{i=1}^N g(d_i|\boldsymbol{x}_i, \boldsymbol{\theta})$ is the decision gain over the entire dataset and $\boldsymbol{d} = [d_1, \dots, d_N]^T$ is the decision vector. In the long-tailed setting, we naturally want the model to fit the testing distribution. Therefore, the integrated gain used in our method is defined as:[4]

$$G(\boldsymbol{d}) := \mathbb{E}_{(\boldsymbol{x}_1, y_1), \dots, (\boldsymbol{x}_N, y_N) \sim p_{\text{test}}(\boldsymbol{x}, y)} \mathbb{E}_{\boldsymbol{\theta} \sim p(\boldsymbol{\theta}|\mathcal{D})} G(\boldsymbol{d}|\boldsymbol{X}, \boldsymbol{\theta}). \tag{4}$$

This function serves as our training objective, integrating both the data distribution and the decision gain within a single function. To use this objective in long-tailed data, there are four main challenges left to address: **i.** How to compute the decision gain $g(d|\boldsymbol{x}, \boldsymbol{\theta})$? (Section 4.1) **ii.** How to handle the long-tailed distributed training data? (Section 4.2) **iii.** How to learn the intractable posterior distribution $p(\boldsymbol{\theta}|\mathcal{D})$? (Section 4.3) **iv.** How to make optimal decisions during testing? (Section 4.4)

### 4.1    How to Design the Decision Gain?

To quantify the utility of making a decision, we define the decision gain as:

$$g(d|\boldsymbol{x}, \boldsymbol{\theta}) := \prod_{y'} p(y'|\boldsymbol{x}, \boldsymbol{\theta})^{u(y', d)}. \tag{5}$$

---

[4]The notion $p_{\text{test}}(\boldsymbol{x}, y)$ is the testing distribution and will be discussed in Section 4.2.

Here, $u(y', d)$ refers to the *utility function* which scores the decision $d$ when the true label is $y'$. The utility values play a critical role in re-weighting the likelihoods of different decisions to prioritize particular mispredictions and allow us to encode human knowledge and expertise tailored to specific tasks.

Eq. 5 is different from previous work like Cobb et al. (2018), which uses $g'(d|\boldsymbol{x}, \boldsymbol{\theta}) := \sum_{y'} U_{y',d} \cdot p(y'|\boldsymbol{x}, \boldsymbol{\theta})$. Both definitions achieve the goal of averaging the predictive probability $p(y|\boldsymbol{x}, \boldsymbol{\theta})$ weighted by the utility values. However, Eq. 5 has two advantages: i) Stability: Eq. 5 is more stable for training. After taking the logarithm, Eq. 5 becomes $\sum_{y'} U_{y',d(\boldsymbol{x})} \cdot \log p(y'|\boldsymbol{x}, \boldsymbol{\theta})$ which is a weighted average of the log probabilities, while Cobb et al. (2018) becomes $\log \sum_{y'} U_{y',d(\boldsymbol{x})} \cdot p(y'|\boldsymbol{x}, \boldsymbol{\theta})$, which is a log of weighted average of the probabilities and not commonly used in classification problems. ii) Flexibility: Eq. 5 allows for more general and flexible utility values whereas Cobb et al. (2018) requires the utility matrix $\boldsymbol{U}$ to be positive definite (otherwise we may not be able to compute the logarithm). Due to these reasons, we use Eq. 5.

The theoretical foundation of utility function has been comprehensively studied in Robert et al. (2007); Berger (2013). For example, Chapter 2.2 of Robert et al. (2007) guarantees the existence of utility functions with rational decision-makers. Note that our design of the decision gain differs from previous works Lacoste-Julien et al. (2011); Cobb et al. (2018) for the benefits of objective optimization. A detailed discussion can be found in Appendix A.

Once we have defined the form of the decision gain $g(d|\boldsymbol{x}, \boldsymbol{\theta})$, the next step is designing the utility function $u(y', d)$. For classification tasks, we can employ a utility matrix $\boldsymbol{U}$:

$$\boldsymbol{U} := \begin{bmatrix} u(0,0) & u(0,1) & \cdots & u(0,|\mathcal{Y}|) \\ u(1,0) & u(1,1) & \cdots & u(1,|\mathcal{Y}|) \\ \vdots & \vdots & \ddots & \vdots \\ u(|\mathcal{Y}|,0) & u(|\mathcal{Y}|,1) & \cdots & u(|\mathcal{Y}|,|\mathcal{Y}|) \end{bmatrix}, \tag{6}$$

where $|\mathcal{Y}|$ is the number of classes and $U_{ij} = u(y = i, d = j)$ is the utility score assigned to the case of predicting class $i$ as $j$. The utility matrix serves as the task-specific knowledge and remains unchanged during training and testing. Users can assign negative values to discourage certain mispredictions and positive values to encourage the desired ones, which reflects their knowledge about specific tasks. To illustrate how to design utility matrices for different long-tailed problem settings, we provide practical examples in Fig. 1. For an in-depth discussion on utility designing, please refer to Chapter 2 of Robert et al. (2007).

**Standard Classification.** The standard long-tailed classification can be regarded as a special case in our framework, where the overall accuracy is the most decisive metric in evaluation. In this case, the focus lies solely on determining whether the decision aligns with the ground truth (i.e., $y = d$). As shown in Fig. 1(a), a simple one-hot utility can be defined by $u(y', d) = \mathbb{1}\{y' = d\}$, which corresponds to the standard accuracy metric.

**Tail-sensitive Classification.** Tail-class samples often have high importance due to their scarcity. Mispredictions from tail classes to head classes usually induce severe consequences in real-world tasks, such as activity recognition (Rahman et al., 2021) and medical images (Yang et al., 2022). In these domains, dangerous actions and illnesses are often scarce yet profoundly harmful if neglected. Besides, the lack of training samples in tail classes has been empirically proved to be the bottleneck of classification performance (Li et al., 2022). Therefore, the ratio of false head samples in evaluation can often reflect a model's real-world potential. To this end, a tail-sensitive utility matrix can be defined by adding extra penalties on false head mispredictions, as shown in Fig. 1(b). The tail-sensitive utility matrix encourages the model to predict uncertain samples as tail rather than head, without affecting correct predictions made with confidence.

**Class/Meta-class-sensitive Classification.** In certain applications where semantic differences exist between different categories, preventing mispredictions between specific (meta) classes becomes crucial, regardless of their class probabilities within the long-tailed distribution. For example, object detection systems at airports must avoid misclassifying birds as planes to ensure flight safety (Shi et al., 2021), and autonomous driving systems must prevent misclassifying mammals on the road as vehicles (both are meta-classes including various animal and vehicle types) to ensure driving safety (Yudin et al., 2019). In these contexts, we

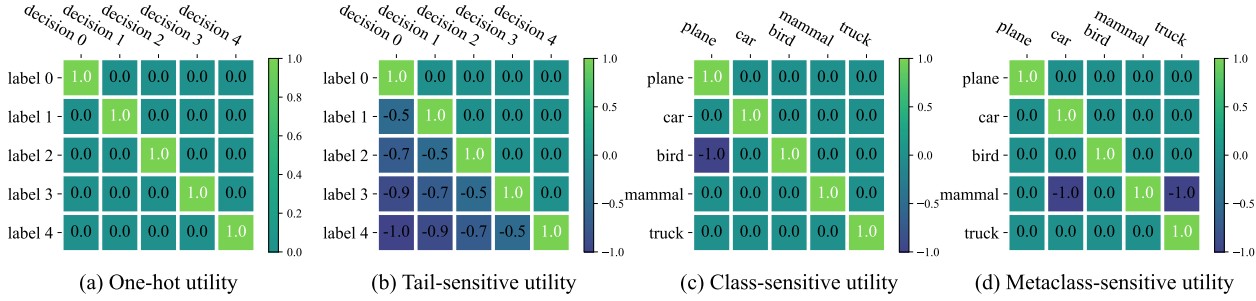

Figure 1: Examples of utility matrices, designed for (a) standard and (b) tail-sensitive classifications, along with (c) class-sensitive classification for bird-plane misprediction and (d) meta-class-sensitive classification for mammal-vehicle misprediction. The entries in the matrix reflect the risk levels ($-1$ is the most risky) and can be flexibly assigned based on task requirements.

provide two examples in Fig. 1(c) and (d) to illustrate how utility values can be assigned to prevent the specific mispredictions that are of primary concern.

These utility matrices enable us to customize decision-making based on the particular objectives and challenges in different long-tailed classification scenarios, improving the reliability and flexibility in real-world applications.

## 4.2 How to handle the Distribution Shift?

In long-tailed classification, there exists a distribution shift between the training (long-tailed) and testing (uniform) sets. We denote the data distributions of the training and testing sets as $p_{\text{train}}(\boldsymbol{x}, y)$ and $p_{\text{test}}(\boldsymbol{x}, y)$ respectively. Since all models are aimed to perform well on the testing set, the data distribution in Eq. 4 should be the testing distribution $p_{\text{test}}(\boldsymbol{x}, y)$. To address the discrepancy between the training and testing distributions, importance sampling (Kloek & Van Dijk, 1978) is adopted:

$$
\mathbb{E}_{(\boldsymbol{x},y)\sim p_{\text{test}}(\boldsymbol{x},y)}\Psi(\boldsymbol{x}, y) = \int p_{\text{test}}(\boldsymbol{x}, y)\Psi(\boldsymbol{x}, y)d(\boldsymbol{x}, y) = \int p_{\text{train}}(\boldsymbol{x}, y)\frac{p_{\text{test}}(\boldsymbol{x}, y)}{p_{\text{train}}(\boldsymbol{x}, y)}\Psi(\boldsymbol{x}, y)d(\boldsymbol{x}, y)
$$
$$
= \mathbb{E}_{(\boldsymbol{x},y)\sim p_{\text{train}}(\boldsymbol{x},y)}\frac{p_{\text{test}}(\boldsymbol{x}, y)}{p_{\text{train}}(\boldsymbol{x}, y)}\Psi(\boldsymbol{x}, y),
\tag{7}
$$

where $\Psi(\boldsymbol{x}, y)$ denote any possible function, which will be specified in Section 4.3. The ratio $p_{\text{test}}(\boldsymbol{x}, y)/p_{\text{train}}(\boldsymbol{x}, y)$ explicitly shows the discrepancy between training and testing data. One common assumption in long-tailed data is that distributional differences only exist between classes (Hong et al., 2021). Formally, this assumption can be expressed as follows:

**Assumption 1** (Intra-class Consistency). *The distributional differences only exist between classes. Given a fixed class label, the data distributions are the same for training and testing data: $p_{train}(\boldsymbol{x}|y) = p_{test}(\boldsymbol{x}|y)$.*

Based on Assumption 1, the discrepancy ratio can be further simplified:

$$
\frac{p_{\text{test}}(\boldsymbol{x}, y)}{p_{\text{train}}(\boldsymbol{x}, y)} = \frac{p_{\text{test}}(y)p_{\text{test}}(\boldsymbol{x}|y)}{p_{\text{train}}(y)p_{\text{train}}(\boldsymbol{x}|y)} = \frac{p_{\text{test}}(y)}{p_{\text{train}}(y)},
\tag{8}
$$

which only depends on the class probabilities of training and testing data. Given that the testing set is assumed to be uniform, the probability $p_{\text{test}}(y)$ would be a constant, and thus the ratio is equivalent to:

$$
\frac{p_{\text{test}}(y)}{p_{\text{train}}(y)} \propto \frac{1}{p_{\text{train}}(y)} \propto \frac{1}{f(n_y)},
\tag{9}
$$

where $f(\cdot)$ is an increasing function and $n_y$ refers to the number of samples in the class $y$. We introduce the notion of $f(n_y)$ because the class probability only depends on the number of samples in this class. Many

existing re-weighting methods in long-tailed classification can be regarded as special instances of $f(\cdot)$. For example, $f(n_y) = n_y^\gamma$ is the most conventional choice with a sensitivity factor $\gamma$ to control the importance of head classes (Huang et al., 2016; Wang et al., 2017; Pan et al., 2021); $f(n_y) = (1 - \beta^{n_y})/(1 - \beta)$ is the effective number which considers data overlap (Cui et al., 2019). A detailed analysis of the choice of $f(\cdot)$ in RF-DLC is conducted in Section 5.5.

### 4.3  How to Learn the Posterior Distribution?

The posterior distribution $p(\boldsymbol{\theta}|\mathcal{D})$ in the integrated gain (Eq. 4) is generally intractable. To learn the posterior, we use a variational distribution $q(\boldsymbol{\theta})$ to approximate the true posterior $p(\boldsymbol{\theta}|\mathcal{D})$. Specifically, we find a tractable lower bound for the logarithm of the integrated gain:

$$
\begin{aligned}
\log G(\boldsymbol{d} = \boldsymbol{Y}) &= \log \mathbb{E}_{(\boldsymbol{x}_1,y_1),\dots,(\boldsymbol{x}_N,y_N)\sim p_{\text{test}}(\boldsymbol{x},y)} \mathbb{E}_{\boldsymbol{\theta}\sim p(\boldsymbol{\theta}|\mathcal{D})} G(\boldsymbol{d} = \boldsymbol{Y}|\boldsymbol{X},\boldsymbol{\theta}) \\
&\geq \sum_{i=1}^{N} \mathbb{E}_{(\boldsymbol{x}_i,y_i)\sim p_{\text{test}}(\boldsymbol{x},y)} \mathbb{E}_{\boldsymbol{\theta}\sim q(\boldsymbol{\theta})} \left[ \sum_{y'} U_{y',d_i=y_i} \cdot \log p(y'|\boldsymbol{x}_i,\boldsymbol{\theta}) + \log p(y_i|\boldsymbol{x}_i,\boldsymbol{\theta}) \right] - KL(q(\boldsymbol{\theta})||p(\boldsymbol{\theta})) + C \\
&\approx \sum_{i=1}^{N} \mathbb{E}_{\boldsymbol{\theta}\sim q(\boldsymbol{\theta})} \frac{1}{f(n_{y_i})} \left[ \sum_{y'} U_{y',d_i=y_i} \cdot \log p(y'|\boldsymbol{x}_i,\boldsymbol{\theta}) + \log p(y_i|\boldsymbol{x}_i,\boldsymbol{\theta}) \right] - KL(q(\boldsymbol{\theta})||p(\boldsymbol{\theta})) + C \\
&:= L(q, \boldsymbol{d} = \boldsymbol{Y}),
\end{aligned}
\tag{10}
$$

where $C = -\log p(\boldsymbol{Y}|\boldsymbol{X})$ is a constant. Here, the second equation is obtained using Jensen's inequality and the third equation is obtained using the importance sampling discussed in Section 4.2 as well as one-sample Monte Carlo (MC) approximation of the expectation over data points $(\boldsymbol{x}_i, y_i)$. The detailed derivation of this lower bound is similar to previous works (Lacoste-Julien et al., 2011; Cobb et al., 2018) and is proved in Appendix D. In the lower bound, we set $d_i = y_i$ since the true label is considered to be the optimal decision at the training stage. By maximizing $L(q, \boldsymbol{d} = \boldsymbol{Y})$ w.r.t. $q$, we obtain a good approximation for the posterior $p(\boldsymbol{\theta}|\mathcal{D})$.

**Relationship with Standard Variational Inference.** On the relationship between our method and standard variational inference (Jordan et al., 1999), a clear similarity is the notion of variational distribution $q(\boldsymbol{\theta})$, which is expected to approximate a posterior distribution. However, due to the distributional shift in long-tailed data, we should also specify whether the posterior distribution is based on training or testing distribution, which are denoted as $p_{\text{train}}(\boldsymbol{\theta}|\mathcal{D})$ and $p_{\text{test}}(\boldsymbol{\theta}|\mathcal{D})$ respectively. We show below that the lower bound $L$ contains the objective for standard variational inference:

$$
L(q, \boldsymbol{d} = \boldsymbol{Y}) \approx \mathbb{E}_{(\boldsymbol{x}_1,y_1),\dots,(\boldsymbol{x}_N,y_N)\sim p_{\text{test}}(\boldsymbol{x},y)} \mathbb{E}_{\boldsymbol{\theta}\sim q(\boldsymbol{\theta})} \log G(\boldsymbol{d} = \boldsymbol{Y}|\boldsymbol{X},\boldsymbol{\theta}) - KL(q(\boldsymbol{\theta})||p_{\text{test}}(\boldsymbol{\theta}|\mathcal{D})),
\tag{11}
$$

where the approximation is due to the use of MC approximation. If further assuming the utility matrix to be one-hot (i.e., $U_{y,d} = \mathbb{1}\{y = d\}$), the equation can be further simplified as:

$$
L(q, \boldsymbol{d} = \boldsymbol{Y}) \approx -KL(q(\boldsymbol{\theta})||p_{\text{test}}(\boldsymbol{\theta}|\mathcal{D})).
\tag{12}
$$

The proof is provided in Appendix E.

**Particle-based Variational Distribution.** To pursue the efficiency of posterior inference, we construct the variational distribution using particles (Liu & Wang, 2016; D'Angelo & Fortuin, 2021):

$$
q(\boldsymbol{\theta}) = \sum_{j=1}^{M} w_j \cdot \delta(\boldsymbol{\theta} - \boldsymbol{\theta}_j),
\tag{13}
$$

where $\{w_j\}_{j=1}^{M}$ are normalized weights which hold $\sum_{j=1}^{M} w_j = 1$, and $\delta(\cdot)$ is the Dirac delta function. The "particles" $\{\boldsymbol{\theta}_j\}_{j=1}^{M}$ are implemented through ensemble models. Empirical studies have previously explored ensemble methods in the context of long-tailed data (Wang et al., 2020; Li et al., 2022). Our framework

provides a theoretical foundation for these approaches in long-tailed problems: due to the scarcity of tailed data, there is not enough evidence to support a single solution, leading to many equally good solutions (which give complementary predictions) in the loss landscape. Therefore, estimating the full posterior distribution is essential to get a comprehensive characterization of the solution space. Particle optimization reduces the cost of Bayesian inference and is more efficient than variational inference (Blundell et al., 2015) and Markov chain Monte Carlo (MCMC) (Brooks et al., 2011), especially on the high-dimensional and multimodal deep neural network posteriors.

**Repulsive Regularization.** The integrated gain optimization in Eq. 10 includes a regularization term $KL(q(\boldsymbol{\theta})||p(\boldsymbol{\theta}))$, which keeps $q$ to be close to the prior $p(\boldsymbol{\theta})$. If we assume the prior $p(\boldsymbol{\theta})$ to be a Gaussian distribution, the regularization can be extended to:

$$KL(q(\boldsymbol{\theta})||p(\boldsymbol{\theta})) = \lambda \int_{\Theta} ||\boldsymbol{\theta}||^2 \cdot q(\boldsymbol{\theta})d\boldsymbol{\theta} + \int_{\Theta} q(\boldsymbol{\theta}) \log q(\boldsymbol{\theta})d\boldsymbol{\theta} = \frac{\lambda}{M} \sum_{j=1}^{M} ||\boldsymbol{\theta}_j||^2 - H(\boldsymbol{\theta}), \tag{14}$$

where $\lambda$ is a constant, $\Theta$ is the parameter space and $H(\boldsymbol{\theta})$ is the entropy of $\boldsymbol{\theta}$. The first term ($L_2$-regularization) prevents the model from over-fitting and the second term (entropy) applies a *repulsive force* to individual models to promote their diversity, pushing the particles away from each other (Liu & Wang, 2016; D'Angelo & Fortuin, 2021). To make the entropy term computable, we introduce a simple approximation: $H(\boldsymbol{\theta}) \approx \frac{1}{2} \log |\hat{\Sigma}_{\boldsymbol{\theta}}|$, where $\hat{\Sigma}_{\boldsymbol{\theta}}$ is the covariance matrix estimated by particles. Other entropy approximations can also be used. By the technique of SWAG-diagonal covariance (Maddox et al., 2019), the covariance matrix can then be directly computed by: $\hat{\Sigma}_{\boldsymbol{\theta}} = diag(\overline{\boldsymbol{\theta}^2} - \overline{\boldsymbol{\theta}}^2)$. Overall, the regularization term is a combination of $L_2$ weight decay and the repulsive force and the final form of this regularization term is:

$$KL(q(\boldsymbol{\theta})||p(\boldsymbol{\theta})) \approx \frac{\lambda}{M} \sum_{j=1}^{M} ||\boldsymbol{\theta}_j||^2 - \frac{1}{2} \sum_{k} \log (\overline{\boldsymbol{\theta}^2} - \overline{\boldsymbol{\theta}}^2)_k, \tag{15}$$

which is different from existing diversity regularization used in long-tailed classification (Wang et al., 2020; Li et al., 2022). The repulsive regularization in our method is naturally derived from the integrated gain optimization, with strong theoretical motivation.

### 4.4 How to make optimal decisions during testing?

Following the standard Bayesian Decision Theory, we maximize the logarithm of the posterior expected gain $G(d^\star|\boldsymbol{x}^*, \mathcal{D})$ in Eq.(2) for each testing input $\boldsymbol{x}^*$ to obtain optimal decisions:

$$d^\star = \arg \max_{d} \log G(d|\boldsymbol{x}^*, \mathcal{D}) \approx \arg \max_{d} \sum_{j=1}^{M} \sum_{y'} U_{y',d} \cdot \log p(y'|\boldsymbol{x}^*, \boldsymbol{\theta}_j). \tag{16}$$

where the approximation is due to $q(\boldsymbol{\theta}) \approx p(\boldsymbol{\theta}|\mathcal{D})$. Importantly, for symmetric utility matrices (e.g., one-hot utility in Fig. 1(a)), Eq. 16 can be further simplified to $d^\star \approx \arg \max_{d} \sum_{j=1}^{M} \log p(d|\boldsymbol{x}, \boldsymbol{\theta}_j)$, which aligns with the standard ensemble models.

In summary, the integrated gain enables our method to simultaneously consider the posterior distribution, decision-making (utility matrix), and data distribution. It provides a principled way to address more realistic problems on the long-tailed data. The proposed RF-DLC is summarized in Algorithm 1.

## 5 Experiments

In this section, we show the experimental results to demonstrate the effectiveness of our method. We use CIFAR10/100-LT (Cui et al., 2019), ImageNet-LT (Liu et al., 2019b), and iNaturalist (Van Horn et al., 2018) as the long-tailed datasets, and compare our method with multiple long-tailed baselines. Detailed implementation is summarized in Appendix F.

---

**Algorithm 1:** RF-DLC

---

**Inputs:** Dataset $\mathcal{D} = \{\boldsymbol{X}, \boldsymbol{Y}\} = \{(\boldsymbol{x}_i, y_i)\}_{i=1}^N$, initial particles $\{\boldsymbol{\theta}_j\}_{j=1}^M$, utility matrix $\boldsymbol{U}$, the step size $\eta$;
**Results:** Final particles $\{\boldsymbol{\theta}_j^*\}_{j=1}^M$;

**for** *each iteration* **do**
    $\boldsymbol{X}_\Xi, \boldsymbol{Y}_\Xi \leftarrow$ A mini-batch sampled from $\mathcal{D}$;
    $L_\Xi(q, \boldsymbol{d} = \boldsymbol{Y}_\Xi) \leftarrow$ Loss computed by Eq. 10;
    **for** $j = 1, ..., M$ **do**
        $\boldsymbol{\theta}_j \leftarrow \boldsymbol{\theta}_j - \eta \cdot \nabla_{\boldsymbol{\theta}_j} L_\Xi$ ;                             /* Update $q(\boldsymbol{\theta})$ */
    **end**
**end**

---

Table 1: False Head Rate evaluation for tail-sensitive long-tailed classification (%). Three datasets and three tail region settings are considered. Our method consistently outperforms all baselines across all settings.

| Dataset | Tail Ratio | CE | CB Loss (Cui et al., 2019) | LDAM (Cao et al., 2019) | RIDE (Wang et al., 2020) | TLC (Li et al., 2022) | RF-DLC |
|---|---|---|---|---|---|---|---|
| CIFAR10-LT | 25% | $21.10 \pm 0.43$ | $14.84 \pm 0.93$ | $10.05 \pm 1.01$ | $8.94 \pm 0.66$ | $10.42 \pm 0.64$ | $\mathbf{4.99 \pm 0.32}$ |
| | 50% | $37.87 \pm 0.57$ | $27.98 \pm 1.44$ | $19.64 \pm 1.66$ | $17.80 \pm 1.39$ | $20.27 \pm 0.77$ | $\mathbf{11.76 \pm 0.29}$ |
| | 75% | $48.75 \pm 1.39$ | $33.93 \pm 1.60$ | $21.37 \pm 2.10$ | $19.77 \pm 3.20$ | $22.24 \pm 1.53$ | $\mathbf{11.01 \pm 1.28}$ |
| | average | $35.91 \pm 0.54$ | $25.58 \pm 1.27$ | $17.02 \pm 1.56$ | $15.50 \pm 1.68$ | $17.64 \pm 0.93$ | $\mathbf{9.25 \pm 0.49}$ |
| CIFAR100-LT | 25% | $45.53 \pm 1.54$ | $24.88 \pm 0.34$ | $21.22 \pm 0.99$ | $18.83 \pm 0.70$ | $21.18 \pm 0.54$ | $\mathbf{15.39 \pm 0.57}$ |
| | 50% | $73.03 \pm 1.59$ | $48.41 \pm 1.24$ | $43.04 \pm 1.18$ | $39.50 \pm 1.53$ | $41.15 \pm 0.55$ | $\mathbf{31.34 \pm 0.55}$ |
| | 75% | $91.30 \pm 1.24$ | $74.38 \pm 1.47$ | $65.62 \pm 1.31$ | $62.01 \pm 2.70$ | $61.34 \pm 1.03$ | $\mathbf{49.51 \pm 1.45}$ |
| | average | $69.95 \pm 1.40$ | $49.22 \pm 0.83$ | $43.29 \pm 1.04$ | $40.11 \pm 1.62$ | $41.22 \pm 0.55$ | $\mathbf{32.08 \pm 0.78}$ |
| ImageNet-LT | 25% | $3.99 \pm 0.08$ | $3.66 \pm 0.17$ | $4.17 \pm 0.19$ | $3.62 \pm 0.18$ | $3.47 \pm 0.13$ | $\mathbf{2.70 \pm 0.09}$ |
| | 50% | $12.77 \pm 0.29$ | $11.80 \pm 0.12$ | $12.73 \pm 0.28$ | $11.42 \pm 0.27$ | $11.49 \pm 0.13$ | $\mathbf{9.68 \pm 0.25}$ |
| | 75% | $30.99 \pm 0.40$ | $29.39 \pm 0.28$ | $29.90 \pm 0.41$ | $26.92 \pm 0.33$ | $27.12 \pm 0.13$ | $\mathbf{24.42 \pm 0.19}$ |
| | average | $15.92 \pm 0.17$ | $14.95 \pm 0.15$ | $15.60 \pm 0.21$ | $13.99 \pm 0.24$ | $14.03 \pm 0.09$ | $\mathbf{12.27 \pm 0.08}$ |

### 5.1 Tail-sensitive Long-tailed Classification with False Head Rate

As discussed in Section 4.1, mispredictions from tail classes to head classes generally pose higher risks and quantifying the likelihood of such occurrences is crucial. Inspired by the false positive rate, we define the *False Head Rate* (FHR) as follows:

$$FHR = \frac{|\mathcal{P}_{\text{head}} \cap \mathcal{G}_{\text{tail}}|}{|\mathcal{G}_{\text{tail}}|}, \tag{17}$$

where $\mathcal{G}_{\text{tail}}$ is the set of samples that are labeled as tail classes and $\mathcal{P}_{\text{head}}$ is the set of samples that are predicted as head classes. To consider different tail regions, we select the last 25%, 50%, and 75% classes as tail classes, respectively. We apply the tail-sensitive utility in Fig. 1(b) to our method. From Table 1, we observe substantial improvements over all baselines across all settings, especially on the relatively small CIFAR datasets, which means that the False Head Rate is more challenging on small datasets.

### 5.2 Class-sensitive Long-tailed Classification

We further try the (meta) class sensitive cases on CIFAR10-LT, including bird-plane detection (Shi et al., 2021) and vehicle-mammal detection (Yudin et al., 2019), as discussed in Section 4.1. The utility matrices used for these two tasks are Fig. 1 (c) and (d) respectively, and their full formats are shown in Appendix C. We also design two corresponding metrics for accurate evaluation, and list the results in Table 2. We again observe that our method improves significantly on the metrics with negligible drops in standard accuracy. These real-world tasks show the importance of taking the decision loss into account and also demonstrate the flexibility of our method which is compatible with different utilities, leading to better performance for different types of tasks.

Table 2: Class-sensitive long-tailed classification on CIFAR10-LT, comparing the baseline one-hot utility with class/metaclass-sensitive utility. The specifically designed utilities can effectively improve tailored metrics with negligible drops in standard overall accuracy.

| Task | Metric | Evaluation Score (%) ↓ | ACC (%) ↑ |
|---|---|---|---|
| bird-plane detection | $R_{\text{plane}} = \frac{|\mathcal{P}_{\text{plane}} \cap \mathcal{G}_{\text{bird}}|}{|\mathcal{G}_{\text{bird}}|}$ | $5.10 \rightarrow 3.80 \ (-25.5\%)$ | $84.11 \rightarrow 83.84 \ (-0.3\%)$ |
| vehicle-mammal detection | $R_{\text{mammal}} = \frac{|\mathcal{P}_{\text{vehicle}} \cap \mathcal{G}_{\text{mammal}}|}{|\mathcal{G}_{\text{vehicle}}|}$ | $1.40 \rightarrow 0.35 \ (-75.5\%)$ | $84.11 \rightarrow 83.83 \ (-0.3\%)$ |

Table 3: Top-1 overall accuracy and tail-class accuracy on standard long-tailed classification (%). Our method outperforms all baselines on both metrics.

| Method | CIFAR10-LT | | CIFAR100-LT | | ImageNet-LT | |
|---|---|---|---|---|---|---|
| | All | Tail | All | Tail | All | Tail |
| LA† (Menon et al., 2020) | 77.67 | - | 43.89 | - | 55.11 | - |
| ACE† (Cai et al., 2021) | 81.2 | - | 49.4 | 23.5 | 54.7 | - |
| SRepr† (Nam et al., 2023) | $82.06 \pm 0.01$ | - | $47.81 \pm 0.02$ | $23.31 \pm 0.11$ | $52.12 \pm 0.06$ | $32.14 \pm 0.41^*$ |
| CE | $73.65 \pm 0.39$ | $58.51 \pm 0.62$ | $38.82 \pm 0.52$ | $10.62 \pm 1.23$ | $47.80 \pm 0.15$ | $44.03 \pm 0.24$ |
| CB Loss (Cui et al., 2019) | $77.62 \pm 0.69$ | $68.73 \pm 1.52$ | $42.24 \pm 0.41$ | $20.50 \pm 0.51$ | $51.70 \pm 0.25$ | $48.29 \pm 0.41$ |
| LDAM (Cao et al., 2019) | $80.63 \pm 0.69$ | $77.14 \pm 1.61$ | $43.13 \pm 0.67$ | $23.50 \pm 1.28$ | $51.04 \pm 0.21$ | $47.21 \pm 0.22$ |
| RIDE (Wang et al., 2020) | $83.11 \pm 0.52$ | $79.62 \pm 1.56$ | $48.99 \pm 0.44$ | $28.78 \pm 1.52$ | $54.32 \pm 0.54$ | $50.74 \pm 0.62$ |
| TLC (Li et al., 2022) | $79.70 \pm 0.65$ | $76.39 \pm 0.98$ | $48.75 \pm 0.16$ | $28.40 \pm 0.72$ | $55.03 \pm 0.34$ | $51.56 \pm 0.35$ |
| RF-DLC | $\mathbf{83.75 \pm 0.17}$ | $\mathbf{82.33 \pm 1.16}$ | $\mathbf{50.24 \pm 0.70}$ | $\mathbf{30.34 \pm 1.49}$ | $\mathbf{55.73 \pm 0.17}$ | $\mathbf{51.98 \pm 0.40}$ |

## 5.3 Standard Long-tailed Classification

We evaluate the overall accuracy and tail-class accuracy. The results are shown in Table 3[5], where † means the results are directly copied from the original papers and ∗ is due to a different setting of tail classes[6]. We apply the one-hot utility in our method. In particular, Our method significantly outperforms other baselines on the crucial tailed data. The results on iNaturalist are summarized in appendix G. These results demonstrate the reliability of our method in standard long-tailed classification.

## 5.4 Uncertainty Estimation

In our method, the predictive uncertainty can be naturally obtained by the entropy of predictive distribution (Malinin & Gales, 2018). For the compared uncertainty estimation algorithms, MCP is a trivial baseline that obtains uncertainty scores from the maximum value of softmax distribution (Hendrycks & Gimpel, 2017); evidential uncertainty is rooted in the subjective logic (Jsang, 2018), and is introduced to long-tailed classification by Li et al. (2022). We evaluate the three uncertainty algorithms in Table 4. Our Bayesian predictive uncertainty outperforms the other two and has a remarkable advantage on the ECE metric, demonstrating the superiority of using principled Bayesian uncertainty quantification.

## 5.5 Ablation Studies

**Effect of Utility.** The effect of tail-sensitive utility is shown in Table 5. We compare the one-hot and tail-sensitive utilities in terms of False Head Rate and standard overall accuracy. By applying the tail-sensitive utility, the performances on FHR can be significantly improved $(18.00\%)$ with a negligible drop in standard accuracy $(0.04\%)$. Besides, we also observe that the performance is relatively robust to the specific values of utility as long as the sign is correct. For example, changing $-1$ to $-0.8$ in the utility matrices in Fig. 1 will not significantly affect the performance of our method (ACC: from 50.24 to 50.34, FHR: from 32.08 to 32.85, on CIFAR100-LT).

**Forms of Class Probability.** We compare five different forms of $f(n_y)$ in terms of standard overall accuracy in Table 6 and Fig. 2a. We also analyze the weight values (i.e., $1/f(n_y)$) and their growth rates

---

[5]Note that the classification evaluation is based on balanced test data.

[6]The number of tail classes of Nam et al. (2023) on ImageNet is less than ours, and thus their tail-class accuracy would naturally be much lower.

Table 4: Calibration evaluation of different uncertainty algorithms. The Bayesian predictive uncertainty adopted in our method outperforms other algorithms.

| Uncertainty Algorithm | CIFAR10-LT | | CIFAR100-LT | | ImageNet-LT | |
| --- | --- | --- | --- | --- | --- | --- |
| | AUC (%) ↑ | ECE (%) ↓ | AUC (%) ↑ | ECE (%) ↓ | AUC (%) ↑ | ECE (%) ↓ |
| MCP (Hendrycks & Gimpel, 2017) | $79.98 \pm 0.10$ | $14.33 \pm 0.37$ | $80.48 \pm 0.51$ | $23.75 \pm 0.51$ | $84.02 \pm 0.24$ | $18.35 \pm 0.12$ |
| Evidential (Li et al., 2022) | $83.20 \pm 0.59$ | $13.24 \pm 0.55$ | $77.37 \pm 0.33$ | $21.64 \pm 0.47$ | $81.45 \pm 0.13$ | $15.29 \pm 0.12$ |
| Bayesian (RF-DLC) | $\mathbf{86.83 \pm 0.68}$ | $\mathbf{9.84 \pm 0.17}$ | $\mathbf{81.24 \pm 0.25}$ | $\mathbf{10.35 \pm 0.28}$ | $\mathbf{84.45 \pm 0.09}$ | $\mathbf{8.72 \pm 0.13}$ |

Table 5: Comparison of different utilities in terms of False Head Rate on CIFAR100-LT. The tail-sensitive utility can effectively improve FHR with a negligible drop in standard overall accuracy.

| Utility | FHR (%) @tail ratio ↓ | | | | Better (%) | ACC (%) ↑ | Worse (%) |
| --- | --- | --- | --- | --- | --- | --- | --- |
| | 25% | 50% | 75% | average | | | |
| one-hot | $18.55 \pm 0.38$ | $38.62 \pm 0.62$ | $60.17 \pm 1.48$ | $39.12 \pm 0.72$ | 18.00 | $\mathbf{49.91 \pm 0.33}$ | 0.04 |
| tail-sensitive | $\mathbf{15.39 \pm 0.57}$ | $\mathbf{31.34 \pm 0.55}$ | $\mathbf{49.51 \pm 1.45}$ | $\mathbf{32.08 \pm 0.78}$ | | $49.89 \pm 0.19$ | |

between the first and the last class across different forms. We find that as the growth rate becomes larger, ACC will be better accordingly, which suggests a high level of class imbalance in the dataset.

Fig. 2a shows similar results on the relationship between growth rate and the tail-class accuracy. As the growth rate becomes larger, the tail and med accuracy will both become significantly better despite the slight drop in head accuracy, which is consistent with the improvement in overall accuracy. Based on these results, we suggest using $f(n_y) = n_y$ in general.

**Number of Particles.** Generally, using more individual models will induce better performances. However, we also need to balance the performance with the computational cost. We visualize accuracy under different numbers of particles in Fig. 2b. The error bars are scaled to be within two standard deviations. The accuracy curves are all logarithm-like and the accuracy improvement is hardly noticeable for more than 6 particles. However, the computational cost is increasing at a linear speed. Therefore, we recommend using no more than six particles in practice for a desirable performance-cost trade-off.

**Repulsive Force.** We also evaluate the effectiveness of the repulsive force. The repulsive force effectively pushes the particle-based variational distribution to the target posterior and avoids collapsing into the same solution. Therefore, with the repulsive force, better predictive distributions can be learned, and thus better predictive uncertainty can be obtained. Besides, the repulsive force can also improve the overall accuracy by promoting the diversity of particles. We show the ablation study on repulsive force in Table 7.

Table 6: Comparison of different forms of class probabilities. Top-1 standard accuracy evaluated on CIFAR100-LT. The linear $f(n_y)$ is the most suitable form in terms of standard accuracy.

| Form of $f(n_y)$ | | Weight Value | | | ACC (%) ↑ |
|---|---|---|---|---|---|
| | | first class | last class | growth (%) | |
| linear (Wang et al., 2017) | $n_y$ | 0.0020 | 0.1667 | 8250 | **50.17 ± 0.25** |
| effective number (Cui et al., 2019) | $(1 - \beta^{n_y})/(1 - \beta)$ | 0.0023 | 0.1669 | 7297 | 49.90 ± 0.36 |
| sqrt (Pan et al., 2021) | $\sqrt{n_y}$ | 0.0447 | 0.4082 | 814 | 47.03 ± 0.30 |
| log | $\log n_y$ | 0.1609 | 0.5581 | 247 | 45.26 ± 0.51 |
| constant | $C$ | 1.0000 | 1.0000 | 0 | 43.27 ± 0.30 |

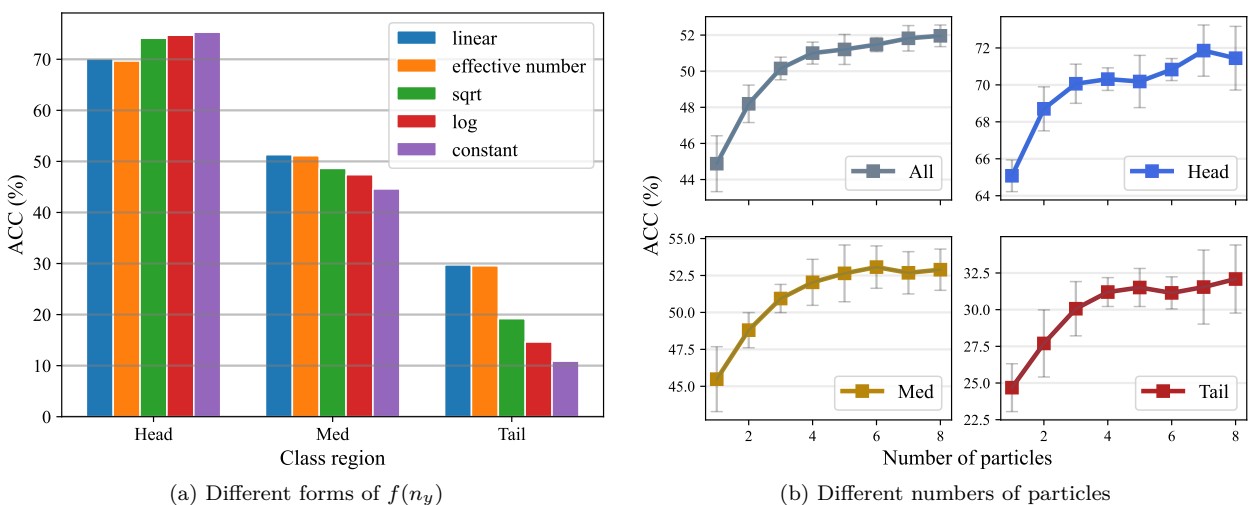

(a) Different forms of $f(n_y)$

(b) Different numbers of particles

Figure 2: Ablation studies on CIFAR100-LT. Comparing (a) different forms of class probabilities and (b) varying numbers of particles in terms of overall accuracy and the accuracy of 3 class regions.

Table 7: The effect of repulsive force, compared by calibration on CIFAR100-LT. Applying the repulsive force will induce better predictive uncertainty.

| Repulsive Force | AUC (%) ↑ | ECE (%) ↓ | ACC (%) ↑ |
|---|---|---|---|
| ✓ | **81.24 ± 0.25** | **10.35 ± 0.28** | **50.24 ± 0.70** |
| ✗ | 75.94 ± 0.56 | 13.40 ± 0.80 | 50.15 ± 0.41 |

## 6 Conclusion and Limitations

This paper proposes Making **R**eliable and **F**lexible **D**ecisions in **L**ong-tailed **C**lassification (RF-DLC) to address decision-making in realistic long-tailed problems. We focus on the problem of asymmetric misprediction cost in real-world long-tailed classification, and derive a novel decision-making framework from Bayesian Decision Theory. Specifically, we introduce the integrated gain to naturally incorporate the distributional shift in long-tailed data, and leverage the utility matrix to make flexible decisions for various task settings. In empirical evaluations, we propose a new False Head Rate metric to quantify the particular type of misprediction from tail classes to head classes, along with standard classification and uncertainty estimation experiments. The evaluation results demonstrate the superiority of our method on both standard and decision-critical long-tailed classifications.

However, we believe there are still some limitations for future developments. We list a few limitations below:

**Long-tailed Regression.**   We have not explored long-tailed problems in regression, where the distribution of targets can also be highly imbalanced. However, with adjustments to the decision gain, we believe our framework can be adapted for regression tasks as well.

**Dataset Shift.**   We have not accounted for general dataset shift scenarios, such as out-of-distribution data, where the assumption of semantically identical training and testing sets becomes invalid. Another example is the distribution of testing data. If it is no longer assumed to be uniform, the discrepancy ratio $p_{\text{test}}(\boldsymbol{x}, y)/p_{\text{train}}(\boldsymbol{x}, y)$ will not be expressed as $1/f(n_y)$, but rather in a more general form.

## Broader Impact Statement

As a general framework for long-tailed classification, our method is eligible for handling imbalanced data in many real-world applications. For example, the strategy adopted in our method can promote fairness and improve performance especially for the underrepresented demographic groups. Besides, the use of utility functions allows for considering specific groups and raising their importance. We believe that adapting our method to realistic social problems is an interesting and important direction. We will investigate this direction in our future works.

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

## A   The Design of Decision Gain

This section is moved to the main body.

As mentioned in the paper, we design the decision gain to be the following:

$$g(d|\boldsymbol{x}, \boldsymbol{\theta}) := \prod_{y'} p(y'|\boldsymbol{x}, \boldsymbol{\theta})^{U_{y',d}}. \tag{18}$$

Our design is different from previous work like Cobb et al. (2018), which uses

$$g'(d|\boldsymbol{x}, \boldsymbol{\theta}) := \sum_{y'} U_{y',d} \cdot p(y'|\boldsymbol{x}, \boldsymbol{\theta}). \tag{19}$$

Both definitions achieve the goal of averaging the predictive probability $p(y|\boldsymbol{x}, \boldsymbol{\theta})$ weighted by the utility values. However, our design has two advantages: i) Stability: Eq. 18 is more stable for training. After taking the logarithm, Eq. 18 becomes $\sum_{y'} U_{y',d(\boldsymbol{x})} \cdot \log p(y'|\boldsymbol{x}, \boldsymbol{\theta})$ which is a weighted average of the log probabilities, while Eq. 19 becomes $\log \sum_{y'} U_{y',d(\boldsymbol{x})} \cdot p(y'|\boldsymbol{x}, \boldsymbol{\theta})$, which is a log of weighted average of the probabilities and not commonly used in classification problems. ii) Flexibility: Eq. 18 allows for more general and flexible utility values whereas Eq. 19 requires the utility matrix $\boldsymbol{U}$ to be positive definite (otherwise we may not be able to compute the logarithm in Eq. 19). Due to these reasons, we use Eq. 18 in this paper.

## B   Related Model Architectures

The model architecture of our method is an ensemble of multiple individual models. This architecture belongs to a general type of Bayesian neural networks, called particle-based BNN or particle optimization (Liu & Wang, 2016; D'Angelo & Fortuin, 2021). Generally, the particle-based variational distribution is in the form of $q(\boldsymbol{\theta}) = \sum_{j=1}^{M} w_j \cdot \delta(\boldsymbol{\theta} - \boldsymbol{\theta}_j)$, where $\{w_j\}_{j=1}^{M}$ are normalized weights which hold $\sum_{j=1}^{M} w_j = 1$, and $\delta(\cdot)$ is the Dirac delta function. Each "particle" $\boldsymbol{\theta}_j$ is a deterministic model and provides the predictive probability $p(y|\boldsymbol{x}, \boldsymbol{\theta}_j)$. The particle-based BNN is first studied in Stein variational gradient descent (SVGD) (Liu & Wang, 2016) and then explored by Liu et al. (2019a); Korba et al. (2020); D'Angelo & Fortuin (2020). Instead of directly modeling the gradient flow, our framework optimizes the particles through stochastic gradient descent (SGD), with repulsive force induced by the integrated gain objective. Compared to existing particle optimization, our method is easy and cheap to implement, which is especially beneficial for large-scale deep learning.

Similar architectures like the "multi-expert" models have been explored by previous long-tailed classification methods (Xiang et al., 2020; Wang et al., 2020; Li et al., 2022). They usually combine several individual classifiers with a shared encoder to obtain better generalization performances (Lakshminarayanan et al., 2017; Hu et al., 2022), which is inspired by the observation that multiple i.i.d. initializations are less likely to generate averagely "bad" models (Dietterich, 2000). The particle-based models reduce the cost of Bayesian inference and are more efficient than variational inference and Markov chain Monte Carlo (MCMC), especially on high-dimensional and multimodal distributions. Besides, the computational cost of our method can be further reduced by leveraging recent techniques, such as partially being Bayesian in model architectures (Kristiadi et al., 2020).

## C   Full Utility Matrices for Class/Metaclass-sensitive Utility

We show the full utility matrices of class-sensitive and metaclass-sensitive utilities in Fig. 3, which is based on the 10 categories of CIFAR10-LT (Cui et al., 2019).

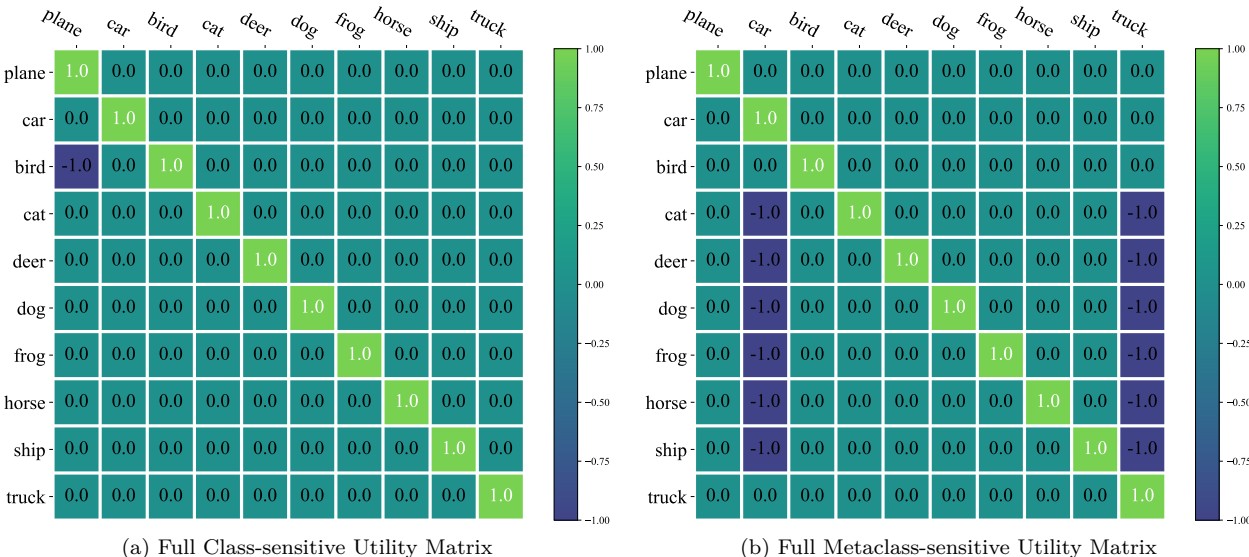

(a) Full Class-sensitive Utility Matrix             (b) Full Metaclass-sensitive Utility Matrix

Figure 3: Full utility matrix configurations on CIFAR10-LT. The two matrices are designed to particularly avoid certain misprediction types (bird-place and mammal-vehicle mispredictions respectively).

## D    Derivation of Eq. 10

The following proof is based on the assumption that both training and testing data are semantically identical[7], and the only difference lies in class probabilities ($p_{\text{train}}(y)$ and $p_{\text{test}}(y)$).

*Proof.* We denote the dataset as $\mathcal{D} = \{\boldsymbol{X}, \boldsymbol{Y}\} = \{(\boldsymbol{x}_i, y_i)\}_{i=1}^N$, where each pair is drawn from the same data distribution. Then the logarithm of integrated gain would be:

$$
\begin{aligned}
\log G(\boldsymbol{d}) &= \log \mathbb{E}_{(\boldsymbol{x}_1, y_1), \ldots, (\boldsymbol{x}_N, y_N) \sim p_{\text{test}}(\boldsymbol{x}, y)} \mathbb{E}_{\boldsymbol{\theta} \sim p(\boldsymbol{\theta}|\mathcal{D})} G(\boldsymbol{d}|\boldsymbol{X}, \boldsymbol{\theta}) \\
&\overset{(a)}{\geq} \mathbb{E}_{(\boldsymbol{x}_1, y_1), \ldots, (\boldsymbol{x}_N, y_N) \sim p_{\text{test}}(\boldsymbol{x}, y)} \log \mathbb{E}_{\boldsymbol{\theta} \sim p(\boldsymbol{\theta}|\mathcal{D})} G(\boldsymbol{d}|\boldsymbol{X}, \boldsymbol{\theta}) \\
&= \mathbb{E}_{(\boldsymbol{x}_1, y_1), \ldots, (\boldsymbol{x}_N, y_N) \sim p_{\text{test}}(\boldsymbol{x}, y)} \log \int_{\boldsymbol{\Theta}} q(\boldsymbol{\theta}) G(\boldsymbol{d}|\boldsymbol{X}, \boldsymbol{\theta}) \frac{p(\boldsymbol{\theta}|\mathcal{D})}{q(\boldsymbol{\theta})} d\boldsymbol{\theta} \\
&\overset{(b)}{\geq} \mathbb{E}_{(\boldsymbol{x}_1, y_1), \ldots, (\boldsymbol{x}_N, y_N) \sim p_{\text{test}}(\boldsymbol{x}, y)} \int_{\boldsymbol{\Theta}} q(\boldsymbol{\theta}) \log \left[ G(\boldsymbol{d}|\boldsymbol{X}, \boldsymbol{\theta}) \frac{p(\boldsymbol{\theta}|\mathcal{D})}{q(\boldsymbol{\theta})} \right] d\boldsymbol{\theta} \\
&:= \mathbb{E}_{(\boldsymbol{x}_1, y_1), \ldots, (\boldsymbol{x}_N, y_N) \sim p_{\text{test}}(\boldsymbol{x}, y)} \Psi(\mathcal{D}).
\end{aligned}
\tag{20}
$$

Here, (a) and (b) are by Jensen's inequality (Jensen, 1906). The $\Psi(\mathcal{D})$ inside the expectation can be further extended to be:

$$
\begin{aligned}
\Psi(\mathcal{D}) &= \int_{\boldsymbol{\Theta}} q(\boldsymbol{\theta}) \log \left[ G(\boldsymbol{d}|\boldsymbol{X}, \boldsymbol{\theta}) \frac{p(\boldsymbol{\theta}|\mathcal{D})}{q(\boldsymbol{\theta})} \right] d\boldsymbol{\theta} \\
&= \int_{\boldsymbol{\Theta}} q(\boldsymbol{\theta}) \log \left[ G(\boldsymbol{d}|\boldsymbol{X}, \boldsymbol{\theta}) \cdot \frac{p(\boldsymbol{\theta})}{q(\boldsymbol{\theta})} \cdot \frac{p(\boldsymbol{Y}|\boldsymbol{X}, \boldsymbol{\theta})}{p(\boldsymbol{Y}|\boldsymbol{X})} \right] d\boldsymbol{\theta} \\
&= \int_{\boldsymbol{\Theta}} q(\boldsymbol{\theta}) \left[ \log \prod_{i=1}^N g(d_i|\boldsymbol{x}_i, \boldsymbol{\theta}) - \log \frac{q(\boldsymbol{\theta})}{p(\boldsymbol{\theta})} + \log \prod_{i=1}^N p(y_i|\boldsymbol{x}_i, \boldsymbol{\theta}) - \log p(\boldsymbol{Y}|\boldsymbol{X}) \right] d\boldsymbol{\theta} \\
&= \sum_{i=1}^N \mathbb{E}_{\boldsymbol{\theta} \sim q(\boldsymbol{\theta})} \left[ \log g(d_i|\boldsymbol{x}_i, \boldsymbol{\theta}) + \log p(y_i|\boldsymbol{x}_i, \boldsymbol{\theta}) \right] - KL(q(\boldsymbol{\theta})||p(\boldsymbol{\theta})) - \log p(\boldsymbol{Y}|\boldsymbol{X}).
\end{aligned}
\tag{21}
$$

---

[7]In contrast to the open-set scenario (Geng et al., 2020), where additional classes may cause testing data to be semantically irrelevant to training data.

Therefore, the ultimate expression of integrated gain is:

$$\log G(\boldsymbol{d}) \geq \mathbb{E}_{(\boldsymbol{x}_1,y_1),...,(\boldsymbol{x}_N,y_N)\sim p_{\text{test}}(\boldsymbol{x},y)}\Psi(\mathcal{D})$$

$$= \sum_{i=1}^{N}\mathbb{E}_{(\boldsymbol{x}_i,y_i)\sim p_{\text{test}}(\boldsymbol{x},y)}\mathbb{E}_{\boldsymbol{\theta}\sim q(\boldsymbol{\theta})}\left[\log g(d_i|\boldsymbol{x}_i,\boldsymbol{\theta}) + \log p(y_i|\boldsymbol{x}_i,\boldsymbol{\theta})\right] - KL(q(\boldsymbol{\theta})||p(\boldsymbol{\theta})) - \log p(\boldsymbol{Y}|\boldsymbol{X})$$

$$= \sum_{i=1}^{N}\mathbb{E}_{(\boldsymbol{x}_i,y_i)\sim p_{\text{train}}(\boldsymbol{x},y)}\mathbb{E}_{\boldsymbol{\theta}\sim q(\boldsymbol{\theta})}\frac{p_{\text{test}}(\boldsymbol{x}_i,y_i)}{p_{\text{train}}(\boldsymbol{x}_i,y_i)}\left[\log g(d_i|\boldsymbol{x}_i,\boldsymbol{\theta}) + \log p(y_i|\boldsymbol{x}_i,\boldsymbol{\theta})\right] - KL(q(\boldsymbol{\theta})||p(\boldsymbol{\theta})) - \log p(\boldsymbol{Y}|\boldsymbol{X})$$

$$\overset{(c)}{\approx} \sum_{i=1}^{N}\mathbb{E}_{\boldsymbol{\theta}\sim q(\boldsymbol{\theta})}\frac{p_{\text{test}}(\boldsymbol{x}_i,y_i)}{p_{\text{train}}(\boldsymbol{x}_i,y_i)}\left[\log g(d_i|\boldsymbol{x}_i,\boldsymbol{\theta}) + \log p(y_i|\boldsymbol{x}_i,\boldsymbol{\theta})\right] - KL(q(\boldsymbol{\theta})||p(\boldsymbol{\theta})) - \log p(\boldsymbol{Y}|\boldsymbol{X})$$

$$\propto \sum_{i=1}^{N}\mathbb{E}_{\boldsymbol{\theta}\sim q(\boldsymbol{\theta})}\frac{1}{f(n_{y_i})}\left[\sum_{y'}U_{y',d_i}\cdot\log p(y'|\boldsymbol{x}_i,\boldsymbol{\theta}) + \log p(y_i|\boldsymbol{x}_i,\boldsymbol{\theta})\right] - KL(q(\boldsymbol{\theta})||p(\boldsymbol{\theta})) - \log p(\boldsymbol{Y}|\boldsymbol{X})$$

$$:= L(q,\boldsymbol{d}).$$

(22)

Here, (c) is by the one-sample MC estimates of the expectation over the data point $(\boldsymbol{x}_i, y_i)$. □

## E Proof of Eq. 11 and Eq. 12

*Proof.* The objective in Eq. 10 can be written as:

$$L(q,\boldsymbol{d}) \approx \sum_{i=1}^{N}\mathbb{E}_{(\boldsymbol{x}_i,y_i)\sim p_{\text{test}}(\boldsymbol{x},y)}\mathbb{E}_{\boldsymbol{\theta}\sim q(\boldsymbol{\theta})}\left[\log g(d_i|\boldsymbol{x}_i,\boldsymbol{\theta}) + \log p(y_i|\boldsymbol{x}_i,\boldsymbol{\theta})\right] - KL(q(\boldsymbol{\theta})||p(\boldsymbol{\theta})) - \log p(\boldsymbol{Y}|\boldsymbol{X})$$

$$= \mathbb{E}_{(\boldsymbol{x}_1,y_1),...,(\boldsymbol{x}_N,y_N)\sim p_{\text{test}}(\boldsymbol{x},y)}\sum_{i=1}^{N}\mathbb{E}_{\boldsymbol{\theta}\sim q(\boldsymbol{\theta})}\left[\log g(d_i|\boldsymbol{x}_i,\boldsymbol{\theta}) + \log p(y_i|\boldsymbol{x}_i,\boldsymbol{\theta})\right] - KL(q(\boldsymbol{\theta})||p(\boldsymbol{\theta})) - \log p(\boldsymbol{Y}|\boldsymbol{X})$$

$$= \mathbb{E}_{(\boldsymbol{x}_1,y_1),...,(\boldsymbol{x}_N,y_N)\sim p_{\text{test}}(\boldsymbol{x},y)}\mathbb{E}_{\boldsymbol{\theta}\sim q(\boldsymbol{\theta})}\left[\log G(\boldsymbol{d}|\boldsymbol{X},\boldsymbol{\theta}) + \log p(\boldsymbol{Y}|\boldsymbol{X},\boldsymbol{\theta})\right] - KL(q(\boldsymbol{\theta})||p(\boldsymbol{\theta})) - \log p(\boldsymbol{Y}|\boldsymbol{X})$$

$$\overset{(a)}{\approx} \int_{\Theta}q(\boldsymbol{\theta})\left[\log\frac{q(\boldsymbol{\theta})}{p(\boldsymbol{\theta})} - \log p_{\text{test}}(\boldsymbol{Y}|\boldsymbol{X},\boldsymbol{\theta}) + \log p(\boldsymbol{Y}|\boldsymbol{X})\right]d\boldsymbol{\theta} + \mathbb{E}_{(\boldsymbol{x}_1,y_1),...,(\boldsymbol{x}_N,y_N)\sim p_{\text{test}}(\boldsymbol{x},y)}\mathbb{E}_{\boldsymbol{\theta}\sim q(\boldsymbol{\theta})}\log G(\boldsymbol{d}|\boldsymbol{X},\boldsymbol{\theta})$$

$$= -KL(q(\boldsymbol{\theta})||p_{\text{test}}(\boldsymbol{\theta}|\mathcal{D})) + \mathbb{E}_{(\boldsymbol{x}_1,y_1),...,(\boldsymbol{x}_N,y_N)\sim p_{\text{test}}(\boldsymbol{x},y)}\mathbb{E}_{\boldsymbol{\theta}\sim q(\boldsymbol{\theta})}\log G(\boldsymbol{d}|\boldsymbol{X},\boldsymbol{\theta}).$$

(23)

Here, (a) is by $N$ independent one-sample MC approximation over all of the data points. When we choose the one-hot utility matrix $\boldsymbol{U} = \boldsymbol{I}$, the decision gain $G(\boldsymbol{d} = \boldsymbol{Y}|\boldsymbol{X},\boldsymbol{\theta})$ will equal the predictive probability $p(\boldsymbol{Y}|\boldsymbol{X},\boldsymbol{\theta})$, and thus it can be ignored. Therefore, the objective with one-hot utility matrix is:

$$L(q,\boldsymbol{d}) \approx -KL(q(\boldsymbol{\theta})||p_{\text{test}}(\boldsymbol{\theta}|\mathcal{D})). \tag{24}$$

□

## F Implementation Details

We summarize the necessary implementation details in this section for the reproducibility of our method. The hyper-parameter choices are concluded in table 8. The optimal values of those hyper-parameters are determined by grid search. We will release the code after acceptance.

### F.1 Evaluation Protocol

The evaluation protocol consists of standard classification accuracy, three newly designed experiments on the False Head Rate (FHR) along with other two metrics, and calibration experiments on AUC and ECE metrics. Besides, we conduct several ablation studies to evaluate different choices of implementation and the

Table 8: Hyper-parameter configurations.

| Dataset | Base Model | Optimizer | Batch Size | Learning Rate | Training Epochs | Discrepancy Ratio | $\lambda$ | $\tau$ | $\alpha$ |
|---------|-----------|-----------|-----------|--------------|----------------|------------------|-----------|--------|----------|
| CIFAR10-LT | ResNet32 | SGD | 128 | 0.1 | 200 | linear | 5e-4 | 40 | 0.002 |
| CIFAR100-LT | ResNet32 | SGD | 128 | 0.1 | 200 | linear | 5e-4 | 40 | 0.3 |
| ImageNet-LT | ResNet50 | SGD | 256 | 0.1 | 100 | linear | 2e-4 | 20 | 50 |
| iNaturalist | Slim ResNet50 | SGD | 512 | 0.2 | 100 | linear | 2e-4 | 20 | 100 |

effectiveness of components in our method. For all quantitative and visual results, we repeatedly run the experiments five times with random initialization to obtain the averaged results and standard deviations to eliminate random error. We use $f(n_y) = n_y$ unless otherwise specified.

## F.2 Training Objective

We slightly modify the training objective in the code-level implementation to apply two practical tricks for better performance.

**Repulsive force.** We find in experiments that although applying repulsive force can promote the diversity of particles, it will certainly disturb the fine-tuning stage in training, which consequently results in sub-optimal performances by the end of training. To address this issue, we apply an annealing weight to the repulsive force to reduce its effect as the training proceeds:

$$\exp\{-t/\tau\} \cdot \frac{1}{2} \sum_k \log{(\overline{\boldsymbol{\theta}^2} - \overline{\boldsymbol{\theta}}^2)_k}, \tag{25}$$

where $t$ refers to the epoch and $\tau$ is a stride factor that controls the decay of annealing weight. With the annealing weight, the repulsive force will push particles away at the beginning of training, and gradually become negligible at the end of training.

**Utility matrix.** Although the utility matrices in Fig. 1 are designed to address the many realistic problem settings, they will also affect the accuracy of the classification task. Therefore, the utility term in Eq. 10 needs re-scaling so that its negative effect on the accuracy is controllable:

$$\log p(y|\boldsymbol{x}, \boldsymbol{\theta}) + \frac{1}{\alpha} \cdot \sum_{y'} U_{y',d} \cdot \log p(y'|\boldsymbol{x}, \boldsymbol{\theta}), \tag{26}$$

where $\alpha$ is the scaling factor. We can adjust the value of $\alpha$ to carefully control the effect of the utility term, which will bring us significant improvement on the False Head Rate (along with other metrics) with an acceptable accuracy drop.

## F.3 Discussion on Computational Cost

We run all experiments on an NVIDIA RTX A6000 GPU (49 GB) and do not need multiple GPUs for one model. The model architecture follows RIDE (Wang et al., 2020) and TLC (Li et al., 2022), in which the first few layers in neural networks are shared among all particles. Therefore, the computational cost of our method is comparable to existing ensemble models. Besides, compared with gradient-flow-based BNN like D'Angelo & Fortuin (2021), which typically uses 20 particles, our model is far more efficient with no more than 5 particles.

# G Additional Experimental Results

## G.1 Full Experimental Results on Classification

We list the full experimental results of top-1 accuracy in Table 9, including the results on iNaturalist (Van Horn et al., 2018). Classes are equally split into three class regions (head, med and tail). For

example, there are 33, 33 and 34 classes respectively in the head, med and tail regions of CIFAR100-LT. The results on iNaturalist are obtained by a single run due to the large size of the dataset. We use a slim version of ResNet50 for all baselines due to GPU memory limitation. Our method successfully outperforms all other baselines especially on the tailed classes.

Table 9: Full top-1 accuracy results (%). † means the results are directly copied from the original paper.

| Dataset | Method | All | Head | Med | Tail |
|---------|--------|-----|------|-----|------|
| CIFAR10-LT | LA† | 77.67 | - | - | - |
| | ACE† | 81.2 | - | - | - |
| | SRepr† | 82.06 ± 0.01 | - | - | - |
| | CE | 73.65 ± 0.39 | **93.22 ± 0.26** | 74.27 ± 0.42 | 58.51 ± 0.62 |
| | CB Loss | 77.62 ± 0.69 | 91.70 ± 0.57 | 75.41 ± 0.76 | 68.73 ± 1.52 |
| | LDAM | 80.63 ± 0.69 | 90.03 ± 0.47 | 75.88 ± 0.81 | 77.14 ± 1.61 |
| | RIDE | 83.11 ± 0.52 | 91.49 ± 0.40 | **79.39 ± 0.61** | 79.62 ± 1.56 |
| | TLC | 79.70 ± 0.65 | 89.47 ± 0.33 | 74.33 ± 0.96 | 76.39 ± 0.98 |
| | RF-DLC | **83.75 ± 0.17** | 90.49 ± 0.60 | 78.89 ± 0.87 | **82.33 ± 1.16** |
| CIFAR100-LT | LA† | 43.89 | - | - | - |
| | ACE† | 49.4 | 66.1 | 55.7 | 23.5 |
| | SRepr† | 47.81 ± 0.02 | 66.69 ± 0.01 | 49.91 ± 0.01 | 23.31 ± 0.11 |
| | CE | 38.82 ± 0.52 | 68.30 ± 0.61 | 38.39 ± 0.49 | 10.62 ± 1.23 |
| | CB Loss | 42.24 ± 0.41 | 62.53 ± 0.44 | 44.36 ± 0.96 | 20.50 ± 0.51 |
| | LDAM | 43.13 ± 0.67 | 63.58 ± 0.93 | 42.90 ± 1.03 | 23.50 ± 1.28 |
| | RIDE | 48.99 ± 0.44 | 69.11 ± 0.54 | 49.70 ± 0.59 | 28.78 ± 1.52 |
| | TLC | 48.75 ± 0.16 | 69.43 ± 0.36 | 49.02 ± 0.94 | 28.40 ± 0.72 |
| | RF-DLC | **50.24 ± 0.70** | **69.92 ± 0.77** | **51.07 ± 0.82** | **30.34 ± 1.49** |
| ImageNet-LT | LA† | 51.11 | - | - | - |
| | ACE† | 54.7 | - | - | - |
| | SRepr† | 52.12 ± 0.06 | 62.52 ± 0.26 | 49.44 ± 0.18 | 32.14 ± 0.41 |
| | CE | 47.80 ± 0.15 | 53.46 ± 0.36 | 45.92 ± 0.19 | 44.03 ± 0.24 |
| | CB Loss | 51.70 ± 0.25 | 57.62 ± 0.46 | 49.19 ± 0.21 | 48.29 ± 0.41 |
| | LDAM | 51.04 ± 0.21 | 57.66 ± 0.40 | 48.26 ± 0.19 | 47.21 ± 0.22 |
| | RIDE | 54.32 ± 0.54 | 60.88 ± 0.71 | 51.35 ± 0.44 | 50.74 ± 0.62 |
| | TLC | 55.03 ± 0.34 | 61.19 ± 0.53 | 52.35 ± 0.31 | 51.56 ± 0.35 |
| | RF-DLC | **55.73 ± 0.17** | **62.18 ± 0.28** | **53.06 ± 0.22** | **51.98 ± 0.40** |
| iNaturalist | LA† | 66.36 | - | - | - |
| | ACE† | 72.9 | - | - | - |
| | SRepr† | 70.79 ± 0.17 | 70.70 ± 0.31 | **70.83 ± 0.20** | **70.79 ± 0.17** |
| | CE | 64.17 | 75.28 | 63.00 | 54.22 |
| | LDAM | 66.20 | 74.58 | 64.17 | 59.84 |
| | RIDE | 71.07 | 77.21 | 68.23 | 67.78 |
| | RF-DLC | **71.30** | **78.56** | 67.29 | 68.05 |

## G.2 Comparison with Cost-sensitive Learning Approaches

The canonical approaches in cost-sensitive learning (Elkan, 2001) could not solve the decision-making problem in long-tailed classification effectively due to their lack of a holistic approach that integrates data distribution shifts and decision-making processes. For example, the method in Section 2 of Elkan (2001) does not consider specific error types and data distribution during the training phase, and fails to incorporate the utility matrix during testing. The Bayesian method in Section 4 of Elkan (2001) learns a standard posterior without accounting for data distribution or the utility matrix, applying the latter only during the testing phase.

To further illustrate the advantages of our method, we empirically compare our method with two baselines: (1) the Bayesian method in Elkan (2001) and (2) the naive combination of a long-tailed approach, re-

weighting loss, and the Bayesian method in Elkan (2001). The table below (on CIFAR100-LT, with tail-sensitive utility applied) shows that our method significantly outperforms both baselines in terms of ACC and FHR. This demonstrates the advantages of our method which concurrently addresses long-tailed distributions and decision-making during both training and testing phases in a unified way.

Table 10: Comparison with cost-sensitive learning approaches on CIFAR100-LT. Tail-sensitive utility is applied.

| Method | ACC (%) ↑ | | FHR (%) ↓ | | | |
|---|---|---|---|---|---|---|
| | All | Tail | 25% | 50% | 75% | Avg |
| RF-DLC | 49.92 | 33.74 | 14.92 | 30.22 | 51.80 | 32.31 |
| Elkan (2001) | 43.36 | 24.68 | 25.97 | 41.98 | 52.64 | 40.20 |
| Reweighting Loss + Elkan (2001) | 45.40 | 26.35 | 20.69 | 41.28 | 65.36 | 42.44 |

### G.3 Ablation Study on the Robustness of Utility Values

We conducted the experiment on the robustness of utility values on CIFAR100-LT. The utility matrix used in this experiment is:

$$\boldsymbol{U} := \begin{bmatrix} 1 & 0 & \cdots & 0 \\ u & 1 & \cdots & 0 \\ \vdots & \vdots & \ddots & \vdots \\ u & u & \cdots & 1 \end{bmatrix}. \tag{27}$$

The results, as shown in Table 11, demonstrate that all values of $u$ will lead to improvement in FHR and comparable or better ACC compared with the baseline one-hot matrix (i.e. $u = 0$). These findings indicate that our method's performance is relatively insensitive to variations in utility values (we simply use $-1$ in the paper) and fine-tuning these values can further improve performance.

Table 11: Ablation study on the robustness of utility values on CIFAR100-LT.

| Utility Value | ACC (%) ↑ | | FHR (%) ↓ | | | |
|---|---|---|---|---|---|---|
| | All | Tail | 25% | 50% | 75% | Avg |
| 0 | 49.76 | 30.00 | 19.12 | 38.04 | 60.44 | 39.20 |
| -0.1 | 49.70 | 29.88 | 18.12 | 36.56 | 59.32 | 38.00 |
| -0.2 | 49.09 | 29.38 | 18.16 | 36.34 | 60.28 | 38.26 |
| -0.3 | 49.41 | 29.82 | 17.81 | 35.28 | 58.12 | 37.07 |
| -0.4 | 49.49 | 30.41 | 17.20 | 34.64 | 56.80 | 36.21 |
| -0.5 | 49.90 | 31.47 | 16.71 | 33.52 | 55.72 | 35.32 |
| -0.6 | 49.84 | 31.82 | 16.92 | 33.82 | 54.72 | 35.15 |
| -0.7 | 49.32 | 30.85 | 16.33 | 32.04 | 55.08 | 34.48 |
| -0.8 | 49.66 | 33.00 | 15.91 | 31.32 | 51.60 | 32.94 |
| -0.9 | 49.27 | 32.12 | 15.53 | 30.68 | 51.24 | 32.48 |
| -1 | 49.92 | 33.74 | 14.92 | 30.22 | 51.80 | 32.31 |

