# OpenReview forum: "Making Reliable and Flexible Decisions in Long-tailed Classification"
_TMLR — Rejected by TMLR_

### Review · Reviewer_s1RV · 2023-11-15

**Summary Of Contributions:**

The paper introduces a novel method for long-tailed classification (where class probabilities are heavily imbalanced) based on variational inference and Bayesian decision theory. The method is shown to consistently outperform existing methods on standard datasets.

**Audience:**

Yes

**Broader Impact Concerns:**

The paper does not provide a Broader Impact Statement, and while I don't think that such a statement is strictly necessary, I would appreciate if the authors discussed
(a) potential implications of their method on fairness metrics if classes heavily correlate with demographic groups
(b) whether their method can be adapted to ensure high accuracy on all demographic groups (cf. the notion of min-max fairness)

**Claims And Evidence:**

Yes

**Requested Changes:**

*) Inconsistent bibliography style: there are several different formats for papers published at the same conference
*) Equation (10): log is missing in the first line
*) p_train and p_test are formally introduced in Section 4.2, but are used already before; also, is p = p_train?
*) typo in the sentence before Algorithm 1
*) the star for SRepr in Table 3 should also be in the CIFAR 100-LT column

**Strengths And Weaknesses:**

Strengths:
*) the paper proposes a new method that is shown to work better than existing ones
*) detailed appendix
*) extensive experiments that are comparable to prior work (I am missing a reference to Appendix G in the main body of the paper though)

Weaknesses:
*) some things are hard to understand or require a second reading to become clear (e.g., when the authors say "maximize", they never say w.r.t. which variable; it should be repeated that "overall accuracy" in the experiments is w.r.t. the balanced test distribution)
*) the authors should reconsider which content to put in the appendix and which content into the main body (e.g., Appendix A is crucial to understand the beginning of Section 4.1, but the long discussion about the different options for specifying the utility matrix could be removed without affecting understandability)
*) Is there some sign missing in the definition (5) – the way it is now, a negative value of u (which is assigned “to discourage certain mispredictions”) yields a higher decision gain than a positive value of u?

---

> ### Author Response · Authors · 2024-01-11
>
> Thanks for your constructive comments! Please find our responses to your questions below.
>
> **Q1: Some things are hard to understand or require a second reading to become clear.**
>
> A1: Thanks for pointing out these confusions! We will clarify them in the revised version and add a pointer to Appendix G.
>
> **Q2: Appendix A is crucial to understand the beginning of Section 4.1.**
>
> A2: Thanks for the suggestion! We agree that Appendix A will help readers understand the decision gain. We will reorganize these parts in the revised version.
>
> The discussion of utility choices is an important part of illustrating the flexibility of our method. It directly shows that our method can be easily adapted to various tasks by modifying the utility matrix. This is the reason that we chose to keep this part in the main body.
>
> **Q3: Is there any sign missing in the definition (5)?**
>
> A3: Eq. 5 is correct. When $u$ is positive, larger likelihood $p(y|x,\theta)$ induces larger gains; when $u$ is negative, larger $p(y|x,\theta)$ induces smaller gains. The sign of $u$ only controls whether the decision gain is positively or negatively correlated to the likelihood.
>
> During training, the values of $u$ are fixed and the likelihood $p(y|x,\theta)$ is the variable that contributes to the value of decision gain. The values of $u$ reflect the prior knowledge we have for a specific task, and thus they will not change during training.
>
> **Q4: Requested changes.**
>
> A4: We appreciate your attention to detail in identifying the typos. They will be addressed in the revised version.
>
> $p(x,y)$ in Eq. 3 is a general notation of data distribution, without considering the difference between training and testing distributions. We change it to $p_{test}(x,y)$ in Eq. 4 to emphasize that our target is the testing distribution.
>
> The CIFAR100-LT settings in the experiments of Sepr are the same as ours. Therefore we do not add ‘*’.
>
> **Q5: Broader impact concerns**
>
> A5: We did not include the discussion of broader impact since the datasets used in the experiments are standard benchmarks that typically do not raise such concerns. We acknowledge that, as a general framework for long-tailed classification, our method is eligible for handling imbalanced data of different demographic groups. We believe that our method can promote the fairness of different groups while ensuring high performance. We agree that this is a very interesting and important direction. We will include a Broader Impact Statement in the revised version.

---

### Review · Reviewer_4DHF · 2023-12-29

**Summary Of Contributions:**

In this study, the researchers introduce a novel approach for addressing issues in long-tailed classification. They focus on creating a new loss function grounded in Bayesian decision theory. This function is derived from a gain function, calculated by raising the classification decision's utility to the power of the estimated probability. The function also considers the distribution changes that happen during testing. It operates on two main assumptions: the class distribution at test time shifts to uniform and the invariance of p(x|y) from training to testing. The exact version of this loss function is intractable, involving a posterior dependent on the data. To simplify, the authors employ standard methods to form a variational lower bound, while also taking into account the distribution shifts.

The authors conduct experiments on CIFAR- LT, Imagenet LT, iNaturalist datasets and compare their proposal against some other counterparts from the literature. The authors also conduct a range of ablations to contrast the role of different utility matrices.

**Audience:**

Yes

**Broader Impact Concerns:**

I don't observe any specific ethical concerns that necessitate a broader impact statement in this context. Nevertheless, I would advise the authors to consider adding a section on broader impacts, given that the work addresses imbalanced data issues which are pertinent to fairness considerations.

**Claims And Evidence:**

Yes

**Requested Changes:**

Refer to the weaknesses section for my queries and requested modifications. My primary emphasis is on the necessity for authors to clarify what current methods overlook and how their proposal addresses these shortcomings.

**Strengths And Weaknesses:**

**Strengths**

Building new and creative solutions for imbalanced classification is important. Despite the fact that the problem has been known and studied for a while, developing ways that can overcome limitations of existing works is quite important. In this work, the authors have empirically demonstrated that existing works are far from optimal and their proposal seems to make some impressive progress. I found illustrations such as the one shown in Figure 1 and supporting text quite helpful and insightful.

**Weakness**

Although the proposal appears to outperform other methods in practice, my key concern with the paper is that it does not clearly articulate why the proposed method is effective or superior to existing ones. It specifically falls short in explaining the deficiencies of current approaches and how the new solution successfully overcomes these limitations.


Could the authors consider a basic alternative approach that involves balancing the data according to the various classes? It would be beneficial if they could compare their method against this alternative and clarify why their approach would be advantageous. Providing empirical evidence along with some intuitive or theoretical explanations would be greatly valuable.

My next concern is how is one supposed to design the utility matrix in equation (6). Even if we have some domain knowledge about the dataset, the decision of the values that go in the matrix maybe quite arbitrary.

How much do the results vary in different tables, like in Table 2, where the utility matrix isn't one-hot encoded? It's important to ensure that these utility matrices aren't crafted based on the test loss. For example, regarding the utility matrix shown in Figure 3 of Appendix C, how was it formulated?

Finally, what was the need for proposing FHR and not use standard metrics for imbalance classification only?

---

> ### Author Response · Authors · 2024-01-11
>
> Thanks for your supportive and helpful comments! Please find our responses to your questions below.
>
> **Q1: The deficiencies of current approaches and how the new solution successfully overcomes these limitations**
>
> A1: The deficiencies of previous methods: Existing approaches to long-tailed classification usually focus on improving overall accuracy or accuracy for tail classes. However, they often assume that all misclassifications, regardless of the classes involved, carry equal risk. This leads to unreliable predictions where these methods are prone to particular mistakes that are critical in real-world applications. For example, the probability of misclassifying tailed samples as head samples is very high in existing methods as shown in our experiments. Such inaccuracies can result in severe consequences, such as misclassifying patients (a tail class) as healthy individuals (a head class). We have provided a detailed discussion of this in the second paragraph of the introduction.
>
>
> The superiority of our method and how it overcomes the issue: Our method overcomes the issue by integrating decision-making into long-tailed classification, enabling principled and reliable predictions on long-tailed data under various decision utilities.
>
> To achieve this, our method applies Bayesian Decision Theory to long-tailed classification, along with several innovations, including a new loss objective, an efficient variational optimization strategy, and specifically designed utility matrices.
>
> The key strengths of our method are as follows: 1) it enables reliable decision-making by incorporating utility matrices into the training and inference phases in a principled way. 2) it is flexible and can be adapted to various tasks with diverse metrics. Users can adopt our method to many specific fields by re-designing utility matrices. 2) it is rooted in a principled framework based on Bayesian Decision Theory. The techniques are naturally derived from this theory, ensuring that they are integral to the model's function rather than add-on tricks.
>
> We will emphasize these points in the revised version.
>
> **Q2: Could the authors consider a basic alternative approach that involves balancing the data according to the various classes?**
>
> A2: Thanks for the suggestion! Directly balancing the data is traditional and well-studied [1,2,3], but they fall behind the SOTA of long-tailed classification for many years. For example, [11] shows that resampling methods can achieve ACC 0.341 for CIFAR100-LT and 0.376 for ImageNet-LT, while our method attains a notably higher ACC of 0.502 for CIFAR100-LT and 0.557 for ImageNet-LT. Besides, many recent works [6,7,8,9,10] on long-tailed classification have not included these baselines in their experiments. We will add the above explanation to the revised version.
>
> **Q3: How is one supposed to design the utility matrix?**
>
> A3: In Bayesian Decision Theory, utility matrices are typically human-designed. It directly reflects the designer's knowledge of specific tasks. Please refer to Chapter 2 of [14] for an in-depth discussion on this topic.
>
> In this paper, we follow these established guidelines for utility matrix design. The general idea is: If you encourage classifying class A as class B, then the entry $u(A,B)$ should be a positive value; if you discourage this, then $u(A,B)$ should be a negative value. Empirically, we found that the performance of our method is relatively robust to the specific values of $u$ as long as the sign is correct. For example, changing -1 to -0.8 in the utility matrices in Figure 3 will not significantly affect the performance of our method (ACC: from 50.24 to 50.34, FHR: from 32.08 to 32.85, on CIFAR100-LT).
>
> **Q4: How much do the results vary in different tables where the utility matrix isn't one-hot encoded?**
>
> A4: Table 2 and Table 5 illustrate the variability in experimental results when different utility matrices are applied. Generally, using utility other than one-hot will lead to a slight decrease in the standard accuracy, However, the tailored metrics will be significantly improved. This demonstrates the adaptability of our method to different utility matrices while still preserving a high level of accuracy performance.

---

> > ### Comment · Reviewer_4DHF · 2024-02-03
> > **Response**
> >
> > I thank the authors for their responses.
> >
> > In A1, you state and I quote "The deficiencies of previous methods: Existing approaches to long-tailed classification usually focus on improving overall accuracy or accuracy for tail classes. However, they often assume that all misclassifications, regardless of the classes involved, carry equal risk. ".
> > The literature on cost-sensitive learning can address this issue right as they do not assign equal risk to all misclassifications?
> > My question was to better understand why your method improves over what is already known. My question is also related to Reviewer uR7E's question on cost-sensitive classification. I read your response to them. I still do not see what the approach of "integrated gain" buys that existing cost-sensitive classification approaches don't. Can you take a canonical approach (https://cseweb.ucsd.edu/~elkan/rescale.pdf)? and provide intuition on what they miss?

---

> ### Author Response · Authors · 2024-01-11
>
> **Q5: Regarding the utility matrix shown in Figure 3 of Appendix C, how was it formulated?**
>
> A5: The values in the utility matrices used in our experiments are not crafted based on test loss.
> The two utility matrices in Figure 3 of Appendix C are examples of class-sensitive and metaclass-sensitive cases. Since we know the categories in the test set and want to avoid certain types of misprediction (bird-place and mammal-vehicle mispredictions respectively), those types of mispredictions are set to -1 to discourage such mistakes. The correct predictions are set to be 1 and non-risky mispredictions are set to be 0. The values of 1, 0 and -1 are simply picked to keep the correct sign and we found it worked well in the experiments. As mentioned in A3, we observed that minor variations in the values within the utility matrices (e.g. change -1 to -0.8) will not affect the performance of our method.
>
> In summary, our method is not very sensitive to the actual values in the utility matrix. With minimum requirements of prior knowledge and effort, users can design new utility matrices for new tasks.
>
> **Q6: What was the need for proposing FHR and not using standard metrics for imbalance classification only?**
>
> A6: The intuition of FHR is to measure the probability that tailed samples are misclassified as head samples, which is crucial in real-world applications [4,5]. We propose FHR to see if our method can effectively reduce misclassifying tailed samples as head. FHR can also be viewed as an extension of FPR (False Positive Rate) in the multi-class setting.
>
> As far as we know, most long-tailed papers only use overall and tail accuracy in the evaluation [6,7,8,9,10]. If you are referring to the standard metrics such as ROC, AP, and F-measure, they primarily focus on imbalanced _test_ data to fairly compare the performance. The test data of the long-tailed datasets are mostly uniform [9,10,13]. Therefore, many long-tailed papers use standard accuracy, rather than ROC, AP, and F-measure in the evaluation.
>
> [1] Borderline-smote: a new over-sampling method in imbalanced data sets learning. International conference on intelligent computing, 2005.
>
> [2] Exploratory undersampling for class-imbalance learning. IEEE Transactions on Systems, Man, and Cybernetics, 2008.
>
> [3] Feature space augmentation for long-tailed data. ECCV 2020.
>
> [4] Nurse care activity recognition: A cost sensitive ensemble approach to handle imbalanced class problem in the wild. International Symposium on Wearable Computers, 2021.
>
> [5] Proco: Prototype-aware contrastive learning for long-tailed medical image classification. International Conference on Medical Image Computing and Computer-Assisted Intervention, 2022.
>
> [6] Trustworthy Long-Tailed Classification. CVPR 2022.
>
> [7] Long-tailed Recognition by Routing Diverse Distribution-aware Experts. ICLR 2020.
>
> [8] ACE: Ally Complementary Experts for Solving Long-Tailed Recognition in One-Shot. CVPR 2021.
>
> [9] Class-balanced loss based on effective number of samples. CVPR 2019.
>
> [10] Large-scale long-tailed recognition in an open world. CVPR 2019.
>
> [11] How Re-sampling Helps for Long-Tail Learning? NeurIPS 2023.
>
> [12] The Bayesian Choice: From Decision Theoretic Foundations to Computational Implementation. 2007.
>
> [13] The inaturalist species classification and detection dataset. CVPR 2018.
>
> [14] The Bayesian Choice: From Decision Theoretic Foundations to Computational Implementation. 2007.

---

> > ### Comment · Reviewer_4DHF · 2024-02-03
> > **Response**
> >
> > In A4 and A5, you answer the questions regarding utility matrices and I see similar questions were raised by other reviewer too.
> > You state and I quote "In summary, our method is not very sensitive to the actual values in the utility matrix. With minimum requirements of prior knowledge and effort, users can design new utility matrices for new tasks." I am not really convinced of this.
> > Why don't you carry out a robustness experiment? where you properly perturb these utility matrices and show how the results vary? This would give a much clearer picture on how sensitive the numbers are to these particular choices.

---

> > > ### Author Response · Authors · 2024-02-07
> > >
> > > **Q2: Why don't you carry out a robustness experiment, where you properly perturb these utility matrices and show how the results vary?**
> > >
> > > A2: Thank you for your valuable suggestions. In response, we conducted a new experiment (on CIFAR100-LT). We perturb the utility values (u) of the following utility matrix:
> > > $$
> > > \left[\begin{array}{cc}
> > > 1 & 0 & ... & 0 \\\\
> > > u & 1 & ... & 0 \\\\
> > > ... \\\\
> > > u & u & ... & 1
> > > \end{array}\right]
> > > $$
> > > to see how ACC and FHR change. The results, as shown in the table below, demonstrate that all values of u will lead to improvement in FHR and comparable or better ACC compared with the baseline one-hot matrix (i.e. u=0). These findings indicate that our method's performance is relatively insensitive to variations in utility values (we simply use $-1$ in the paper) and fine-tuning these values can further improve performance. This new experiment is also updated in the paper (Appendix G.3).
> > >
> > > | Utility Value | ACC (%) All | ACC (%) Tail | FHR (%) 25% | FHR (%) 50% | FHR (%) 75% | FHR (%) Avg |
> > > |:---:|:---:|:---:|:---:|:---:|:---:|:---:|
> > > | 0 | 49.76 | 30.00 | 19.12 | 38.04 | 60.44 | 39.20 |
> > > | -0.1 | 49.70 | 29.88 | 18.12 | 36.56 | 59.32 | 38.00 |
> > > | -0.2 | 49.09 | 29.38 | 18.16 | 36.34 | 60.28 | 38.26 |
> > > | -0.3 | 49.41 | 29.82 | 17.81 | 35.28 | 58.12 | 37.07 |
> > > | -0.4 | 49.49 | 30.41 | 17.20 | 34.64 | 56.80 | 36.21 |
> > > | -0.5 | 49.90 | 31.47 | 16.71 | 33.52 | 55.72 | 35.32 |
> > > | -0.6 | 49.84 | 31.82 | 16.92 | 33.82 | 54.72 | 35.15 |
> > > | -0.7 | 49.32 | 30.85 | 16.33 | 32.04 | 55.08 | 34.48 |
> > > | -0.8 | 49.66 | 33.00 | 15.91 | 31.32 | 51.60 | 32.94 |
> > > | -0.9 | 49.27 | 32.12 | 15.53 | 30.68 | 51.24 | 32.48 |
> > > | -1 | 49.92 | 33.74 | 14.92 | 30.22 | 51.80 | 32.31 |

---

> ### Author Response · Authors · 2024-02-07
>
> **Q1: Can you take a canonical approach and provide intuition on what they miss?**
>
> A1: The canonical approaches in [1] could not solve the decision-making problem in long-tailed classification effectively due to their lack of a holistic approach that integrates data distribution shifts and decision-making processes. For example, the method in Section 2 of [1] does not consider specific error types and data distribution during the training phase, and fails to incorporate the utility matrix during testing. The Bayesian method in Section 4 of [1] learns a standard posterior without accounting for data distribution or the utility matrix, applying the latter only during the testing phase.
>
> To further illustrate the advantages of our method, we empirically compare our method with two baselines: (1) the Bayesian method in [1] and (2) the naive combination of a long-tailed approach, re-weighting loss, and the Bayesian method in [1]. The table below (on CIFAR100-LT, with tail-sensitive utility applied) shows that our method significantly outperforms both baselines in terms of ACC and FHR. This demonstrates the advantages of our method which concurrently addresses long-tailed distributions and decision-making during both training and testing phases in a unified way. This new experiment is also updated in the paper (Appendix G.2).
>
> | Method | ACC (%) All | ACC (%) Tail | FHR (%) 25% | FHR (%) 50% | FHR (%) 75% | FHR (%) Avg |
> |---|:---:|:---:|:---:|:---:|:---:|:---:|
> | RF-DLC | 49.92 | 33.74 | 14.92 | 30.22 | 51.80 | 32.31 |
> | [1] | 43.36 | 24.68 | 25.97 | 41.98 | 52.64 | 40.20 |
> | Reweighting Loss + [1] | 45.40 | 26.35 | 20.69 | 41.28 | 65.36 | 42.44 |
>
> Finally, we wish to emphasize that in the task of long-tailed classification, no existing cost-sensitive learning method has been applied. This paper is the first to study decision-making in long-tailed classification, extending its applicability to realistic scenarios with non-uniform risks, alongside extensive demonstrations of what is possible, which is in itself a valuable contribution.
>
> [1] The Foundations of Cost-Sensitive Learning. IJCAI 2001.

---

### Review · Reviewer_uR7E · 2024-01-01

**Summary Of Contributions:**

The paper considers the problem of long-tailed classification where certain classes are underrepresented in the training data. It provides a cost-sensitive classification framework for learning models that perform well under class imbalances. The framework relies on bayesian decision theory and aims to learn models that maximize the expected posterior utility. To make the resulting optimization objective tractable, the paper proposes to solve a variational lower bound. The resulting algorithm (RF-DLC) is computationally efficient. The paper demonstrates the utility of the framework on various benchmark datasets for long-tailed classification. The empirical evidence suggests that the proposed approach outperforms a number of existing approaches for long-tailed classification.

**Audience:**

No

**Claims And Evidence:**

No

**Requested Changes:**

1. __Related works__: The current work needs to be properly placed in the literature. Without this, it is hard to understand the contributions of the work.
2. __Comparison with baselines__: Thorough comparison with relevant baselines is needed to better understand the efficacy of the proposed technique.
3. __Lack of experimental details__: Section 5 in the paper lacks enough details to reproduce the results. For example, the architecture choice and training details are never discussed in the paper. It would be great if the authors provide all the necessary details for their approach and the baselines.
4. __Choice of utility matrix__: How does one design the utility/cost matrix in practice?  How did the authors come up with these matrices in Figure 1? Are the performance gains seen in the experiments robust to the choice of the utility matrix? How do the results change for other choices of utility matrices?

**Strengths And Weaknesses:**

__Cost-Sensitive Classification__: The problem considered in the paper is certainly interesting. But one of the main concerns I have is that the work is not well placed among the literature. There is an entire sub-field of ML called cost-sensitive classification that is focused on making optimal decisions in classification (as opposed to simply optimizing the accuracy). For example, see [1] which is one of the earliest works in this line (also see [2]). Some of these techniques are by now standard and are available in many python packages. One could combine these techniques with existing _class re-weighting_ schemes to tackle long-tailed classification. Unfortunately, these techniques are never discussed; neither are these compared with the proposed technique in the experiments section.

__Standard long-tailed classification__: In Section 5.3 (where one-hot utility matrix is used), the authors demonstrated that the proposed technique has superior performance compared to some baselines. But imbalanced classification is a very popular technique and comparison with state-of-the-art is needed to better understand the utility of the approach. It is not clear why/how the current baselines in Table 3 are selected. Why not compare the proposed technique with distributionally robust optimization based techniques [3] and meta-learning based techniques [4]?

__Assumptions__: The test distribution considered in the problem statement looks a bit contrived. In particular, it is not clear why the test distribution should have equal coverage of all classes. This assumption reduces the applicability of the proposed framework. When does this assumption hold in practice (or where does one encounter such settings)? It would be great if the authors add a discussion on this to the paper.



[1] https://cseweb.ucsd.edu/~elkan/rescale.pdf
[2] https://arxiv.org/pdf/1511.09337.pdf
[3] https://arxiv.org/abs/2306.09222
[4] Meta-weight-net: Learning an explicit mapping for sample weighting.

---

> ### Author Response · Authors · 2024-01-11
>
> Thanks for the insightful comments! Please find our responses to your questions below.
>
> **Q1: Related works on cost-sensitive classification are not discussed.**
>
> A1: Bayesian Decision Theory is one of the main methodologies to solve cost-sensitive classification. For example, both references [9] and [10] that you mentioned discuss Bayesian methods. Specifically, [10] states, “This work focuses on multiclass cost-sensitive classification……generally done by equipping probabilistic classifiers with Bayes decision theory”. Our related works focus on Bayesian Decision Theory which is a subfield of cost-sensitive learning and has a closer connection to this study. We will add additional references within the broader context of cost-sensitive classification in the revised version.
>
> To our knowledge, there is no existing research or method that merges techniques from decision-making with long-tailed classification. We want to highlight that this integration is far from straightforward. The challenge lies in effectively and efficiently managing data distribution shifts while also optimizing decision-making processes. Our method can be viewed as a fusion of Bayesian Decision Theory, a popular framework in cost-sensitive learning, with long-tailed classification. To achieve that, we introduce several innovations, including a new loss objective, an efficient variational optimization strategy, and specifically designed utility matrices.
>
> In summary, we believe this work is well-placed in the literature and the contributions are clear and significant.
>
> **Q2: Comparison with state-of-the-art. Why not compare with distributionally robust optimization-based techniques and meta-learning-based techniques?**
>
> A2: In our experiments, we included a total of eight baselines, all of which are recent methods published between 2019 and 2023. Previous works [1,2,3] have also chosen similar baselines as ours.
>
> Since long-tailed classification is a large field, there exist many different types of methods (e.g., re-weighting loss, logit adjustment, knowledge transfer, meta-learning, etc.). We selected baselines that are all based on re-weighting loss, which is of the same type as our method. This minimizes the difference in frameworks and architectures and ensures a fair and meaningful comparison. Our baseline selection aligns with previous works [1,2,3,4,5,7] which usually focus on one type of baseline in their experiments.
>
> Moreover, our method demonstrates superior performance in accuracy even when compared to the mentioned robust optimization-based [11] and meta-learning-based [12] methods. For example, the empirical result for [11] (ACC: CIFAR10-LT 0.7375, CIFAR100-LT 0.4189) and [12] (ACC: CIFAR10-LT 0.7521, CIFAR100-LT 0.4209) are falling behind our method (ACC: CIFAR10-LT 0.8375, CIFAR100-LT 0.5024).
>
> Finally, competing with the current state-of-the-art in accuracy is not the main focus of this paper. These benchmarks often use many empirical tricks to improve performance. This paper focuses on developing a reliable and flexible learning framework to address decision-making problems in long-tailed classification.
>
> In conclusion, we are confident that the baselines used in our experiments are relevant and compelling to adequately demonstrate the effectiveness of our method.
>
> **Q3:  It is not clear why the test distribution should have equal coverage of all classes.**
>
> A3: Uniform testing distribution is the standard setting for long-tailed classification. Most long-tailed datasets follow this setting, including CIFAR-LT [4], ImageNet-LT [5], and iNaturalist [6]. Many previous works, e.g. [1,2,3,7], also employ uniform testing distribution. Therefore, we believe this assumption is natural and aligns with established norms in the long-tailed classification literature.
>
> Moreover, our method can be extended to the case of non-uniform testing distributions with a minor modification to Equation 9. All we need to do is keeping $p_{test}(y)$ in the numerator and calculating it in the same manner as the denominator. Since the exploration of testing distribution is not the focus of this paper, we leave this interesting extension for future works.
>
> **Q4: The architecture choices and training details are never discussed in the paper.**
>
> A4: Due to the space constraint, we primarily focus on the metrics and evaluation results in Section 5. The model architecture is discussed in Appendix B. The details for training are summarized in Algorithm 1 and Appendix F2. We will move the architecture and training details to the main body in the revised version and will release the code upon acceptance to aid in the reproduction of our results.

---

> > ### Comment · Reviewer_uR7E · 2024-02-08
> > **Related Work**
> >
> > I would like to thank the authors for their response. I looked at the updated paper and the author's response.
> >
> > I agree with the claim that "... there is no existing research or method that merges techniques from decision-making with long-tailed classification."  But this aspect is not coming out clearly in the paper. As of now, the paper over claims the contributions. The way introduction is currently written (with statements like "Existing methods assume that mispredictions between any pair of classes are equally risky, which is often violated in real-world tasks.") makes the reader believe that decision theory for classification is something novel that is introduced in the paper. This is not true.
> >
> > Here is my suggestion: Place the work properly in the literature and clearly bring out the contributions/novelty in the work in the introduction. This will make the reader better appreciate the work.

---

> > > ### Author Response · Authors · 2024-02-08
> > >
> > > **Q1: Place the work properly in the literature and clearly bring out the contributions/novelty in the work in the introduction.**
> > >
> > > A1: Thank you for your valuable feedback and suggestions. We have revised the introduction to more clearly position our method in the literature, specifically clarifying that "existing methods" refer to those in long-tailed classification, rather than general classification.

---

> > ### Comment · Reviewer_uR7E · 2024-02-08
> > **standard long-tailed classification**
> >
> > "For example, the empirical result for [11] (ACC: CIFAR10-LT 0.7375, CIFAR100-LT 0.4189) and [12] (ACC: CIFAR10-LT 0.7521, CIFAR100-LT 0.4209) are falling behind our method (ACC: CIFAR10-LT 0.8375, CIFAR100-LT 0.5024)." -- We cannot read numbers from these papers directly because the settings (model architectures, number of epochs, lr etc..) are quite different.  In fact, the baseline CE loss numbers don't match across these 3 works. So, one should be careful in drawing conclusions.
> >
> > My main point is as follows: if the contribution of the work is to combine decision theory with long-tailed classification, I don't see much value in this experiment. The experiments anyways don't move the SOTA and only seem to divert the attention. I'd recommend moving this experiment to the appendix.

---

> > > ### Author Response · Authors · 2024-02-08
> > >
> > > **Q2: Comparison with [11] and [12].**
> > >
> > > A2: The purpose of comparing our method with standard long-tailed classification methods in [11] and [12] is to show that our method not only significantly improves decision-making but also maintains high performance on standard metrics such as accuracy. We agree that advancing the state-of-the-art in accuracy is not the main aim of this paper and will put this experiment in the appendix.

---

> ### Author Response · Authors · 2024-01-11
>
> **Q5: How does one design the utility/cost matrix in practice?**
>
> A5: In Bayesian Decision Theory, and cost-sensitive learning more broadly, utility/cost matrices are typically human-designed. It directly reflects the designer's knowledge of specific tasks. Please refer to Chapter 2 of [8] for an in-depth discussion on this topic.
>
> In this paper, we follow these established guidelines for utility matrix design. The general idea is: If you encourage classifying class A as class B, then the entry $u(A,B)$ should be a positive value; if you discourage this, then $u(A,B)$ should be a negative value. Empirically, we found that the performance of our method is relatively robust to the specific values of $u$ as long as the sign is correct. For example, changing -1 to -0.8 in the utility matrices in Figure 3 will not significantly affect the performance of our method (ACC: from 50.24 to 50.34, FHR: from 32.08 to 32.85, on CIFAR100-LT).
>
> **Q6: How did the authors come up with these matrices in Figure 1?**
>
> A6: The utility matrices in Fig. 1 are specifically crafted to suit the needs of long-tailed classification. The "one-hot utility" corresponds to the standard accuracy. In contrast, the "tail-sensitive utility" introduces penalties for scenarios where tail classes are misclassified as head classes. Additionally, the other two utility matrices impose penalties on specific types of misclassifications. This design rationale stems from the domain knowledge associated with long-tailed classification, where certain errors are considered more costly than others. Detailed explanations and justifications for these utility matrix designs are provided in Section 4.1 of our paper.
>
> **Q7: Are the performance gains seen in the experiments robust to the choice of the utility matrix? How do the results change for other choices of utility matrices?**
>
> A7: As shown in Table 2 and Table 5, with different utility matrices, the accuracy will only slightly drop compared with one-hot utility. However, the tailored metrics will be significantly improved.
> Furthermore, we observed a low sensitivity to minor variations in the values within the utility matrices as mentioned in A5. This demonstrates the adaptability of our method to different utility matrices while still preserving a high level of accuracy performance.
>
> [1] Trustworthy Long-Tailed Classification. CVPR 2022.
>
> [2] Long-tailed Recognition by Routing Diverse Distribution-aware Experts. ICLR 2020.
>
> [3] ACE: Ally Complementary Experts for Solving Long-Tailed Recognition in One-Shot. CVPR 2021.
>
> [4] Class-balanced loss based on effective number of samples. CVPR 2019.
>
> [5] Large-scale long-tailed recognition in an open world. CVPR 2019.
>
> [6] The inaturalist species classification and detection dataset. CVPR 2018.
>
> [7] Learning imbalanced datasets with label-distribution-aware margin loss. NeurIPS 2019.
>
> [8] The Bayesian Choice: From Decision Theoretic Foundations to Computational Implementation. 2007.
>
> [9] The Foundations of Cost-Sensitive Learning. IJCAI 2001.
>
> [10] Cost-aware Pre-training for Multiclass Cost-sensitive Deep Learning. IJCAI 2016.
>
> [11] Stochastic Re-weighted Gradient Descent via Distributionally Robust Optimization. 2023.
>
> [12] Meta-weight-net: Learning an explicit mapping for sample weighting. NeurIPS 2019.

---

### Author Response · Authors · 2024-01-12
**Summary Response to All Reviewers**

We thank all reviewers for their efforts in enhancing the quality of this paper. Based on the suggestions, we revised the paper to include all requested changes. Particularly,

- We added the correlation with cost-sensitive classification in related works.

- We moved the details about the decision gain from the appendix to the main body.

- We added more explanation on the deficiencies of previous methods, the superiority of our method, and the design of utility matrices.

- We corrected typos and bibliography style, and added a Broader Impact Statement.

We believe that this work makes solid and significant contributions to long-tailed classification. To summarize:

- Our method is the first to integrate optimal decision-making into long-tailed classification. We employ Bayesian Decision Theory to formulate decision-making in this context, offering a unified and principled framework.

- To enable optimal decision-making on long-tailed data, we develop several innovations, including a new objective, an efficient variational optimization strategy, and several utility matrices tailored for different long-tailed scenarios.

- Our method adapts flexibly to diverse tasks by the introduction of the utility function/matrix. It significantly improves decision-making while maintaining or even improving traditional metrics such as accuracy and calibration. We provided comprehensive experiments to demonstrate the effectiveness of our method.

We are confident that we address all your questions and requested changes. We would greatly appreciate it if you would consider our response in your evaluation. Please let us know if you have additional questions we can address.

---

### Decision · Action_Editor_YVAc · 2024-03-25

**Recommendation:** Reject

**Comment:**

The authors develop an interesting approach for dealing with long-tail problems in classification. While the empirical results on some datasets are impressive, the improvements are not consistent between all datasets studied. Furthermore, the positioning of the work relative to prior art in cost sensitive classification is unsatisfactory and the authors do not provide sufficient justification regarding why their approach improve upon prior work. Hence, I believe the paper cannot be accepted in its current form, but I encourage the authors to work on the issues raised and resubmit a significantly revised version of the paper.

**Audience:**

All reviewers agree that the paper could potentially be interesting to some of the audience of TMLR.

**Claims And Evidence:**

While the authors make some interesting contributions, there are the following claims that remain not fully substantiated despite the rebuttal process:

1. The empirical results do not show consistent improvement across all datasets, in particular, for the iNaturalist dataset, the approach reduces to the performance of weak baselines.

2. The authors do not position the work relative to prior work appropriately, in particular, they do not show sufficient evidence of why the approach proposed achieves improvements relative to prior work from cost-sensitive classification (see the discussion with reviewers 4DHF and uR7E in particular).

**Resubmission Of Major Revision:**

The authors may consider submitting a major revision at a later time.